# EvoProto: Evolving Prototypes with Class Similarity for Weakly Supervised Incremental Segmentation

## Abstract

Weakly Supervised Incremental Learning for Semantic Segmentation seeks to segment new classes using only image-level labels, without access to old class data, which challenges the stability-plasticity balance. The absence of pixel-level annotations for new classes and historical data for old classes often leads to ***class overwriting***, where predictions for new classes misclassify or override regions belonging to semantically similar previously learned classes. We observe that such overwriting frequently arises from class confusion, where visually similar classes are entangled due to weak supervision and limited feature discrimination. To address this, we propose **EvoProto**, a framework that explicitly models and mitigates class confusion through the dynamic evolution of learnable class prototypes. We begin by introducing a confusion score that quantifies semantic similarity between new and old classes. The adaptive weight, which is calculated from the confusion score and the CAM-derived predictions following a warm-up phase, facilitates both contrastive prototype learning and prototype-level knowledge distillation, thereby enhancing inter-class distinction during continual updates. An additional activation-based label denoising mechanism is applied to emphasize confident and consistent activations among the noise for more reliable weak supervision. Extensive experiments on the Pascal VOC and COCO benchmarks demonstrate that EvoProto effectively alleviates class overwriting and achieves state-of-the-art performance across various incremental scenarios. The code will be made publicly available.

## 1 Introduction

Semantic segmentation assigns each pixel to a predefined class, and although deep learning has greatly advanced its performance Long et al. (2015); Chen (2014); Chen et al. (2017); Chen (2017), such progress depends heavily on large-scale, densely annotated datasets. To reduce annotation costs, Weakly Supervised Semantic Segmentation (WSSS) methods Ahn & Kwak (2018a); Ru et al. (2022); Yang et al. (2024); Wu et al. (2024) utilize image-level labels to generate pseudo-pixel labels through Class Activation Maps (CAMs) Zhou et al. (2016). While WSSS greatly reduces labeling effort, it typically assumes a fixed class set and is not directly applicable to dynamic environments where new semantic categories emerge over time. To address this limitation, Weakly Incremental Learning for Semantic Segmentation (WILSS) has recently been proposed Cermelli et al. (2022) to incrementally learn new classes using only image-level supervision, while preserving segmentation performance on previously learned ones. This setting combines the challenges of weak supervision and continual learning, making it significantly more difficult than either task alone. Specifically, CAMs—already rough in WSSS—become even more unreliable as classification performance degrades during incremental training.

Despite promising progress, existing WILSS methods Cermelli et al. (2022); Yu et al. (2023a); Si et al. (2024); Liu et al. (2024a) overlook two critical and tightly coupled challenges. First, semantic confusion between visually similar classes, such as cow vs. sheep, becomes more severe due to weak supervision, leading to ambiguous representations. Second, the poor quality of CAM-based pseudo-labels for new classes further amplifies this confusion. Together, these issues cause a phenomenon as ***class overwriting***, where new class predictions incorrectly cover regions that belong to semantically

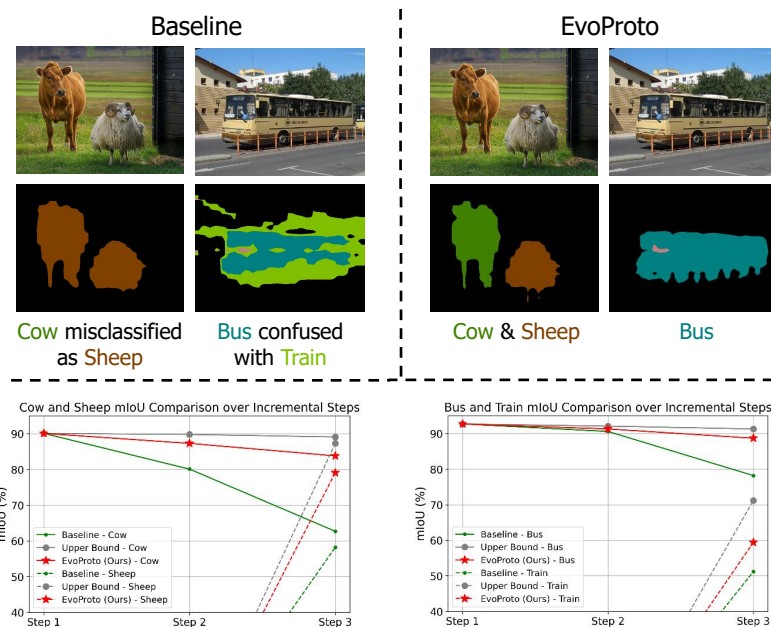

Figure 1: Visualization and quantitative comparison of class overwriting. The images show segmentation results for *cow* vs. *sheep* and *bus* vs. *train*. The **class overwriting** issue occurs in the results of the baseline, where the old classes are partly/totally covered by the similar new classes. EvoProto mitigates the confusion between similar classes and generates the correct prediction result.

similar old classes. This not only harms performance on old categories but also hinders the learning of new ones.

Class overwriting is caused by class confusion (as shown in Fig. 1), *i.e.*, visual and semantic similarity between new and old classes leading to entangled feature representations under weak supervision. To address this, we propose **EvoProto** (**Evo**lving **Proto**types with Class Similarity), a novel WILSS framework that explicitly models and mitigates class overwriting by evolving class-discriminative trainable prototypes via contrastive learning and distillation.

We first define a confusion score to quantify inter-class semantic similarity, computed from CAM-derived predictions after a warm-up phase. This score drives the Confusion-Aware Prototype Optimization by generating two sets of adaptive weights for reweighting: one for contrastive prototype learning, and the other for prototype-level alignment. These reweightings reinforce representation dissimilarity between confusing classes while preserving previously learned knowledge. The prototypes are trainable and dynamically evolve throughout the incremental process. In addition, we observe that weak supervision in WILSS further degrades class separability due to classification inconsistency and pseudo-label noise. To alleviate this, we propose an Activation Based Label Denoising (ALD) mechanism that filters out unreliable activations by identifying confident and consistent CAM channels, resulting in higher-quality pseudo labels for both old and new classes.

In summary, our main contributions are as follows:

- We identify class confusion as the underlying cause of class overwriting in WILSS and propose EvoProto, a framework that addresses this challenge by evolving class-specific prototype representations under weak supervision.

- We address this problem via a prototype evolution strategy, which leverages a confusion score to quantify inter-class semantic similarity. This score generates adaptive weights that guide both contrastive prototype learning and prototype-level alignment, promoting better separation between similar classes while preserving knowledge of old ones.

- We further incorporate an Activation-Based Label Denoising (ALD) mechanism to improve pseudo-label quality under classification degradation. Together with the prototype-based strategy, EvoProto achieves state-of-the-art performance across multiple WILSS benchmarks.

## 2 RELATED WORKS

**Class-Incremental Semantic Segmentation (CISS).** CISS addresses the problem of sequentially learning new classes in semantic segmentation without accessing data from old classes. A major challenge in this setting is *catastrophic forgetting* French (1999); Thrun (1998), where updated representations interfere with previously learned knowledge. Early approaches, such as MiB Cermelli et al. (2020) addressed background shift by dynamically redefining the background class. Subsequent methods adopted strategies like knowledge distillation Li & Hoiem (2017); Baek et al. (2022); Zhang et al. (2022), exemplar replay Cha et al. (2021); Chen et al. (2024a), and pseudo-label refinement Douillard et al. (2021); Cermelli et al. (2023) to mitigate forgetting. However, these methods typically rely on full pixel-level annotations for new classes, which limits their scalability.

**Weakly Supervised Incremental Semantic Segmentation (WILSS).** To reduce annotation cost, WILSS Cermelli et al. (2022); Yu et al. (2023b); Liu et al. (2024b); Si et al. (2024) assumes only image-level labels for new classes while maintaining dense supervision only in the initial stage. This setting is more practical but introduces greater challenges due to the weak nature of the supervision signals. A common issue is *class overwriting*, where noisy or ambiguous pseudo-labels lead to confusion between similar new and old classes. Many WILSS methods adopt a two-stage approach: first generating class-specific pseudo-labels via CAM-based techniques Yu et al. (2023b); Si et al. (2024); Liu et al. (2024b), and then training the segmentation model incrementally. Others enhance pseudo-label quality by incorporating auxiliary cues, such as prompt-based semantic priors Hao et al. (2024). However, these approaches often treat weak supervision and incremental learning separately, making it difficult to fully resolve class interference during continual updates. In contrast to existing methods, our work introduces an integrated framework that jointly considers class similarity and pseudo-label reliability. By estimating inter-class confusion and adaptively reweighting the segmentation loss, our method mitigates class overwriting and improves learning stability under weak supervision.

## 3 PRELIMINARY

**Task Definition**: Incremental semantic segmentation proceeds in sequential steps. At each step $t \in \{0, \ldots, T\}$, the model learns to segment a new set of classes $\boldsymbol{C}^t$, where all class sets are disjoint ($\bigcap_{t=0}^{T} \boldsymbol{C}^t = \varnothing$) and their union forms the complete class set $\mathcal{C} = \bigcup_{t=0}^{T} \boldsymbol{C}^t$. In step $t$, the training data comprises image-label pairs $(\boldsymbol{x}^t, \boldsymbol{y}^t)$, where $\boldsymbol{x}^t$ is the input image and $\boldsymbol{y}^t$ provides dense annotations exclusively for the current classes $\boldsymbol{C}^t$. Annotations for old ($\boldsymbol{C}^{0:t-1}$) and future ($\boldsymbol{C}^{t+1:T}$) classes are unavailable. Based on this, WILSS imposes stricter constraints: only the initial step ($t = 0$) has access to dense labels, while all subsequent steps provide only image-level labels $\boldsymbol{Y}^t$ for the newly introduced classes, with no access to prior data. The objective is to ensure that, after each step $t$, the model can accurately segment all classes seen so far ($\boldsymbol{C}^{0:t}$), effectively integrating new knowledge while mitigating forgetting of previously learned categories.

**CAM based Pseudo Labels**: Given an input image, the feature extractor $\mathcal{F}$ outputs feature maps $\boldsymbol{\mathcal{X}} \in \mathbb{R}^{D_{\text{cls}} \times H \times W}$. Following the common practice Ahn & Kwak (2018a); Wang et al. (2020); Cermelli et al. (2022); Si et al. (2024), Class Activation Maps (CAMs) are computed by taking the inner product between $\boldsymbol{\mathcal{X}}$ and the classifier weights $\boldsymbol{W} \in \mathbb{R}^{|\boldsymbol{C}^{0:t}| \times D_{\text{cls}}}$, *i.e.*, $\text{CAM}(c) = \boldsymbol{W}^{(c)} \cdot \boldsymbol{\mathcal{X}}, \forall c \in \boldsymbol{C}^{0:t}$. The image-level labels $\boldsymbol{Y}$ are then used to suppress absent classes and apply background thresholding, yielding pixel-wise pseudo labels $\boldsymbol{y}_{\text{cam}}$ for new classes. Meanwhile, predictions from the previous model are utilized as auxiliary pseudo labels $\boldsymbol{y}_{\text{pre}}$ for old classes. The final pseudo labels $\hat{\boldsymbol{y}}$ are thus composed by combining the two sources to supervise the segmentation decoder as:

$$\hat{\boldsymbol{y}}(u, v) = \begin{cases} \boldsymbol{y}_{\text{cam}}(u, v), & \text{if } \boldsymbol{y}_{\text{cam}}(u, v) \in \boldsymbol{C}^t \\ \boldsymbol{y}_{\text{pre}}(u, v), & \text{otherwise.} \end{cases} \tag{1}$$

The segmentation decoder is trained with these pseudo labels using a pixel-wise Binary Cross-Entropy loss, encouraging the network to align its predictions with CAM-derived labels for new classes while retaining knowledge of old classes through the previous model's guidance.

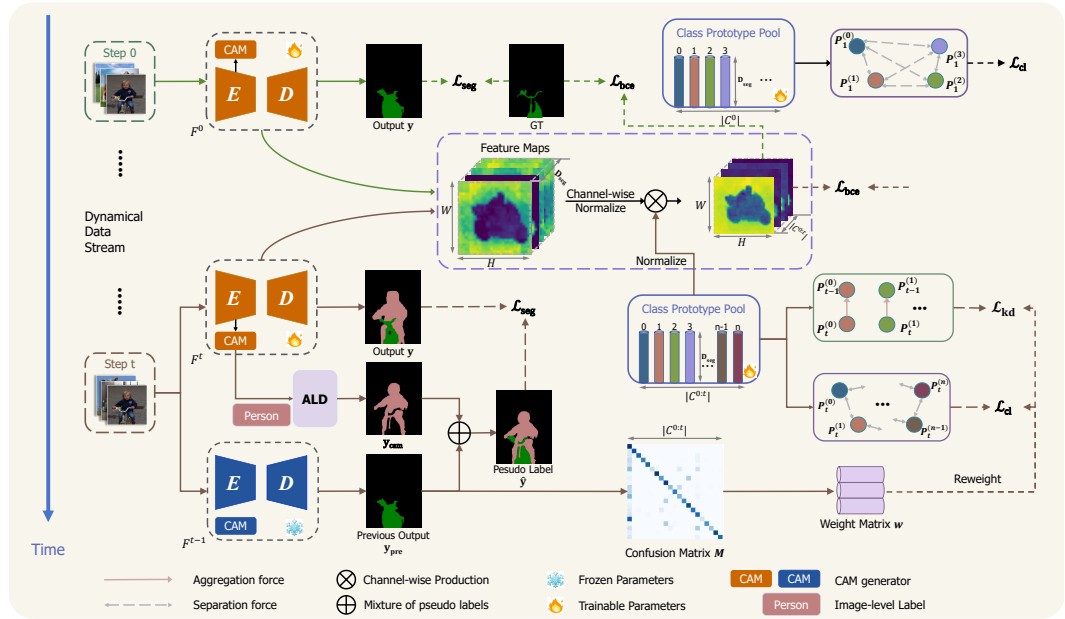

Figure 2: Overall architecture of EvoProto. Given the image and image-level labels, we first estimate a class confusion score to quantify semantic similarity between classes. The Confusion-Aware Prototype Evolving module leverages this score to guide both contrastive prototype learning and prototype-level alignment, promoting class separation and mitigating class overwriting. Meanwhile, the Activation-Based Label Denoising strategy filters low-confidence activations to improve image-level pseudo-label quality under weak supervision.

## 4 PROPOSED METHOD

We propose EvoProto, a confusion-aware framework for WILSS. It mitigates class overwriting by dynamically evolving prototype representations under guidance from class-level semantic similarity. As illustrated in Fig. 2, we first estimate a confusion score to quantify inter-class similarity between classes. This score is used to adaptively weight two prototype-driven objectives: contrastive learning for enhancing class separation, and prototype-level distillation for preserving prior knowledge. To further improve pseudo-label reliability under weak supervision, we introduce an Activation-Based Label Denoising strategy that filters out noisy activations.

### 4.1 CONFUSION-AWARE PROTOTYPE EVOLVING

To address class confusion and overwriting in incremental segmentation, we propose a **Confusion-Aware Prototype Evolving** (**CAPE**) scheme that learns and refines class prototypes over time. Instead of relying on static pixel averages, we introduce learnable prototypes that evolve across training steps, enabling more discriminative and robust class representations. To further guide this evolution, we estimate class-level confusion scores and use them to reweight key training objectives. This reweighting strategy helps mitigate interference between semantically similar categories by adaptively adjusting the strength of prototype separation and knowledge preservation.

### 4.1.1 CLASS-LEVEL CONFUSION SCORING

To measure the semantic similarity between classes, we introduce class-level *confusion scores* during training, which reflect the degree of misclassification induced by feature similarity. These scores serve as unified signals to reweight both the prototype contrastive separation loss and the prototype-level alignment. Given the lack of pixel-level annotations for new classes, we instead use CAM-based pseudo labels obtained at the $K$-th iteration as reference supervision. At the beginning of each subsequent epoch, we construct a confusion matrix by comparing the model predictions $y(u, v)$ with the pseudo labels $\hat{y}(u, v)$ across all pixels $(u, v)$.

For two distinct class $i$ and $j$, the conditional confusion from $i$ to $j$ is defined as

$$B(i \rightarrow j) = \frac{|\{(u,v) \mid y(u,v) = j \wedge \hat{y}(u,v) = i\}|}{|\{(u,v) \mid \hat{y}(u,v) = i\}|}. \tag{2}$$

The symmetric confusion score is then obtained as

$$\boldsymbol{M}_{i,j} = B(i \rightarrow j) + B(j \rightarrow i). \tag{3}$$

The resulting matrix $\boldsymbol{M} \in \mathbb{R}^{|\boldsymbol{C}^{0:t}| \times |\boldsymbol{C}^{0:t}|}$ provides a compact measure of bidirectional confusion between classes. Guided by these scores, the model adaptively balances stability and plasticity: preserving old knowledge when overlap is high, while enforcing stronger separation for easily confused new categories. Furthermore, class-wise confidence refinement is applied to improve the reliability of CAM-based pseudo labels.

### 4.1.2 REWEIGHTED CONTRASTIVE PROTOTYPE LEARNING

Class overwriting often arises between semantically similar classes, causing entire regions to be misclassified or shared ambiguously, which severely degrades segmentation quality. To address this, we propose **Reweighted Contrastive Prototype Learning** (**RCPL**) strategy, which encourages the model to learn more discriminative prototypes through confusion-aware contrastive optimization. Guided by the *confusion score*, the network adaptively controls the degree of separation and alignment among prototypes during training. Instead of using pixel-averaged class representations Kirillov et al. (2019), which are sensitive to noise Michieli & Zanuttigh (2021a) and overlook intra-class variation Hu et al. (2020), we introduce a set of *learnable prototypes* $\boldsymbol{P} \in \mathbb{R}^{|\boldsymbol{C}^{0:t}| \times \boldsymbol{D}_{\text{seg}}}$, where $\boldsymbol{D}_{\text{seg}}$ is the feature dimension of the segmentation decoder. To train these prototypes, we compute the cosine similarity between the segmentation decoder features and the prototypes. The similarity map is then supervised using the pseudo labels via a BCE loss. These learnable prototypes can adaptively fit class-specific distributions, enabling more discriminative and robust representation learning under noisy pseudo-labels and class imbalance.

Once the prototypes $\boldsymbol{P}$ are obtained, we introduce a confusion-weighted contrastive loss to explicitly enforce inter-class separation. A weight matrix $\boldsymbol{w}$ is constructed from the normalized confusion matrix $\boldsymbol{M}$ by applying a centered sigmoid transformation. Since the raw confusion scores are relatively small, they are not directly suitable for use as control weights. Therefore, we employ a normalization function that maps the confusion scores into the range $[-1, 1]$. The rationale for adopting the centered sigmoid, together with a detailed analysis, is provided in the Appendix A.5. Specifically, for a class $i$ and a class $j$, the weight is computed as:

$$w_{i,j} = \begin{cases} 2 \cdot \sigma\left(k \cdot (\boldsymbol{M}_{i,j} - \gamma)\right) - 1, & \text{if } j = CO_i \\ 0, & \text{otherwise.} \end{cases} \tag{4}$$

where $\sigma(\cdot)$ is the sigmoid function, and $k$ and $\gamma$ are scaling and centering hyperparameters, respectively. The index $CO_i$ denotes the counterpart class that is most confused with class $i$. Since our main concern is *class overwriting* introduced when new classes are incrementally added, $i$ and $CO_i$ are always constrained to belong to different phases, i.e., one from the old classes and the other from the new classes. Specifically, if $i$ belongs to an old class, its counterpart $CO_i$ is chosen from the new classes; conversely, if $i$ belongs to a new class, $CO_i$ is selected from the old classes:

$$CO_i = \begin{cases} \arg\max_{j \in \boldsymbol{C}^t} \boldsymbol{M}_{i,j}, & \text{if } i \in \boldsymbol{C}^{0:t-1}, \\ \arg\max_{j \in \boldsymbol{C}^{0:t-1}} \boldsymbol{M}_{i,j}, & \text{if } i \in \boldsymbol{C}^t. \end{cases} \tag{5}$$

This definition ensures that $i$ and $CO_i$ are always selected across the old–new boundary, which is precisely where the *class overwriting* phenomenon most prominently occurs.

Then, each prototype $\boldsymbol{P}_t^{(i)}$ serves as a query, aiming to decrease its similarity with other class prototypes. Finally, the reweighted contrastive prototype loss is defined as:

$$\mathcal{L}_{\text{cl}} = \frac{1}{|\boldsymbol{C}^{0:t}|} \sum_{i=1}^{|\boldsymbol{C}^{0:t}|} \sum_{\substack{j=1 \\ j \neq i}}^{|\boldsymbol{C}^{0:t}|} (1 + \tau \cdot w_{i,j}) \cdot \text{sim}(\boldsymbol{P}_t^{(i)}, \boldsymbol{P}_t^{(j)}), \tag{6}$$

where $\text{sim}(\cdot, \cdot)$ denotes cosine similarity and $\tau$ is a temperature coefficient.

### 4.1.3 PROTOTYPE-LEVEL ALIGNMENT

To balance stability and plasticity during incremental learning, we introduce a **Confusion-Aware Prototype-level Alignment** (**CPA**) that preserves old-class representations while mitigating semantic confusion. The distillation strength is modulated by class-wise confusion scores to prevent over-fitting to ambiguous knowledge. For each old class, given $\boldsymbol{P}_t^{(i)}$ and $\boldsymbol{P}_{t-1}^{(i)}$ as the prototypes of class $i$ from the current and previous models, respectively, we align both prototypes as:

$$\mathcal{L}_{\text{kd}} = \frac{1}{|\boldsymbol{C}^{0:t-1}|} \sum_{i=1}^{|\boldsymbol{C}^{0:t-1}|} (1 - \tau \cdot w_{i,CO_i}) \cdot \left\| \boldsymbol{P}_t^{(i)} - \boldsymbol{P}_{t-1}^{(i)} \right\|_2^2. \tag{7}$$

A lower weight is assigned to highly confused classes to reduce the risk of transferring noisy knowledge, *e.g.*, misinterpreting features of a new class (like *sofa*) as an old one (like *chair*). This adaptive distillation is complemented by the contrastive loss $\mathcal{L}_{\text{cl}}$, which promotes inter-class separation. While $\mathcal{L}_{\text{kd}}$ stabilizes retained representations, $\mathcal{L}_{\text{cl}}$ encourages discriminative feature learning. Together, they enable robust prototype evolution under weak supervision and semantic ambiguity.

### 4.2 ACTIVATION-BASED LABEL DENOISING

In WILSS, CAM-based pseudo labels tend to degrade due to classification decay and the absence of pixel-level annotations, especially for old classes. This leads to noisy supervision and impairs the model's ability to distinguish between semantically similar classes. To alleviate this, we introduce an **Activation-Based Label Denoising** (**ALD**) strategy that improves **image-level** pseudo-label $\boldsymbol{Y}$ reliability by filtering low-confidence activations.

To suppress noisy class activations, we compute a dynamic threshold $thre$ derived from the maximum activation values $\boldsymbol{A}_i$ across the CAM channels of old classes, then CAM is defined with:

$$\boldsymbol{Y}_i^{\text{CAM}} = \begin{cases} 1, & \text{if } \max(\boldsymbol{A}_i) > thre \\ 0, & \text{otherwise} \end{cases} , \quad thre = \frac{1}{|\mathcal{P}| + 1} \sum_{i \in \mathcal{P}} \max(\boldsymbol{A}_i), \tag{8}$$

where $\mathcal{P}$ is the set of classes predicted as present by the old model, and $\boldsymbol{A}_i$ is the activation map of the $i$-th class. The additional $+1$ in the denominator introduces a conservative bias that helps suppress noisy or uncertain predictions—if the predicted labels are reliable, their activations will naturally exceed this slightly lowered threshold; otherwise, low-activation false positives are effectively filtered out. This filtering removes spurious low activations while retaining confident signals.

We then obtain the final image-level pseudo label used for supervision by combining the filtered old-class predictions with ground-truth image-level labels for new classes:

$$\boldsymbol{Y}_{\text{old}} = \boldsymbol{Y}_{\text{old}}^{\text{pred}} \cap \boldsymbol{Y}^{\text{CAM}}, \quad \boldsymbol{Y}_{\text{train}} = [\boldsymbol{Y}_{\text{old}}, \boldsymbol{Y}_{\text{new}}^{\text{gt}}], \tag{9}$$

where '$\cap$' means conjunction of the labels, and $[\cdot, \cdot]$ denotes channel-wise concatenation. The refined pseudo label $\boldsymbol{Y}_{\text{train}}$ is used to supervise the image-level classification head via the binary cross-entropy loss:

$$\mathcal{L}_{\text{cls}} = -\frac{1}{|C^{0:t}|} \sum_{i=1}^{|C^{0:t}|} \left[ \boldsymbol{Y}_{\text{train}}^{(i)} \cdot \log \sigma(\boldsymbol{z}^{(i)}) + (1 - \boldsymbol{Y}_{\text{train}}^{(i)}) \cdot \log(1 - \sigma(\boldsymbol{z}^{(i)})) \right], \tag{10}$$

where $\sigma(\cdot)$ is the sigmoid function, $\boldsymbol{z} \in \mathbb{R}^{|C^{0:t}|}$ is the predicted classification logit vector. This denoising mechanism improves the quality of supervision under weak labels, effectively complementing prototype evolution by reducing activation-level noise and enhancing inter-class separability.

### 4.3 OVERALL OBJECTIVES

We combine multiple loss terms to enhance segmentation accuracy, prototype discrimination, and classification stability under noisy and evolving supervision. The overall objective is

$$\mathcal{L} = \lambda_{\text{cls}} \mathcal{L}_{\text{cls}} + \lambda_{\text{seg}} \mathcal{L}_{\text{seg}} + \lambda_{\text{cl}} \mathcal{L}_{\text{cl}} + \lambda_{\text{kd}} \mathcal{L}_{\text{kd}}, \tag{11}$$

where $\mathcal{L}_{\text{seg}}$ denotes the Binary Cross-Entropy loss with pseudo labels. Note that $\mathcal{L}_{\text{cls}}$ and $\mathcal{L}_{\text{seg}}$ are standard task losses rather than architecture-specific designs.

Table 1: Results on different overlap settings. "P" and "I" denote pixel-level and image-level labels, respectively. Best image-level methods are in **bold**, best pixel-level methods are underlined. FT is a fine-tuning baseline (lower bound), Joint is trained with all classes (upper bound). Entries with "(ViT)" use a Vision Transformer backbone; others use ResNet. †: Methods using the external foundation model SAM Kirillov et al. (2023b).

| Method | Sup | 10-10 VOC | | | 15-5 VOC | | | COCO-to-VOC | | | 10-2 VOC | | | 10-5 VOC | | |
|---|---|---|---|---|---|---|---|---|---|---|---|---|---|---|---|---|
| | | 1-10 | 11-20 | All | 1-15 | 16-20 | All | 1-60 | 61-80 | All | 1-10 | 11-20 | All | 1-10 | 11-20 | All |
| FT | P | 7.8 | 58.9 | 32.1 | 12.5 | 36.9 | 18.3 | 1.9 | 41.7 | 12.7 | - | - | - | - | - | - |
| LWF Li & Hoiem (2017) | P | 70.7 | 63.4 | 67.2 | 67.0 | 41.8 | 61.0 | 36.7 | 49.0 | 40.3 | - | - | - | - | - | - |
| LWF-MC Rebuffi et al. (2017) | P | 53.9 | 43.0 | 48.7 | 59.8 | 22.6 | 51.0 | - | - | - | - | - | - | - | - | - |
| ILT Michieli & Zanuttigh (2019) | P | 70.3 | 61.9 | 66.3 | 69.0 | 46.4 | 63.6 | 37.0 | 43.9 | 39.3 | - | - | - | - | - | - |
| CIL Klingner et al. (2020) | P | 38.4 | 60.0 | 48.7 | 14.9 | 37.3 | 20.2 | - | - | - | - | - | - | - | - | - |
| MiB Cermelli et al. (2020) | P | 70.4 | 63.7 | 67.2 | 75.5 | 49.4 | 69.0 | 34.9 | 47.8 | 38.7 | - | - | - | - | - | - |
| PLOP Douillard et al. (2021) | P | 69.6 | 62.2 | 67.1 | 75.7 | 51.7 | 70.1 | 35.1 | 39.4 | 36.8 | - | - | - | - | - | - |
| SDR Michieli & Zanuttigh (2021b) | P | 70.5 | 63.9 | 67.4 | 75.4 | 52.6 | 69.9 | - | - | - | - | - | - | - | - | - |
| RECALL Maracani et al. (2021) | P | 66.0 | 58.8 | 63.7 | 67.7 | 54.3 | 65.6 | - | - | - | - | - | - | - | - | - |
| ALIFE Oh et al. (2022) | P | 74.1 | 69.8 | 71.9 | 77.2 | 52.5 | 71.3 | - | - | - | 54.8 | 40.7 | 48.1 | 68.3 | 58.8 | 63.8 |
| DKD Baek et al. (2022) | P | 75.2 | 69.6 | 72.5 | 78.8 | 58.2 | 73.9 | - | - | - | 58.7 | 45.8 | 52.6 | 68.8 | 57.6 | 63.4 |
| STAR Chen et al. (2024b) | P | 74.1 | 68.8 | 71.6 | 79.5 | 58.9 | 74.6 | - | - | - | 72.3 | 58.2 | 65.6 | 73.5 | 64.7 | 69.3 |
| BARM Zhang & Gao (2024) | P | 76.2 | 67.8 | 72.2 | 78.5 | 56.3 | 73.2 | - | - | - | 75.1 | 59.7 | 67.8 | 75.7 | 64.8 | 70.5 |
| CAM Zhou et al. (2016) | I | 70.8 | 44.2 | 58.5 | 69.9 | 25.6 | 59.7 | 30.7 | 20.3 | 28.1 | - | - | - | - | - | - |
| SEAM Wang et al. (2020) | I | 67.5 | 55.4 | 62.7 | 68.3 | 31.8 | 60.4 | 31.2 | 28.2 | 30.5 | - | - | - | - | - | - |
| SS Araslanov & Roth (2020) | I | 69.6 | 32.8 | 52.5 | 72.2 | 27.5 | 62.1 | 35.1 | 36.9 | 35.5 | - | - | - | - | - | - |
| EPS Lee et al. (2021) | I | 69.0 | 57.0 | 64.3 | 69.4 | 34.5 | 62.1 | 34.9 | 38.4 | 35.8 | - | - | - | - | - | - |
| WILSON Cermelli et al. (2022) | I | 70.4 | 57.1 | 65.0 | 74.2 | 41.7 | 67.2 | 39.8 | 41.0 | 40.6 | 38.7 | 22.4 | 32.5 | 66.8 | 46.5 | 58.1 |
| Teddy Si et al. (2024) † | I | 71.2 | 59.4 | 66.5 | 77.6 | 51.4 | 72.0 | 40.6 | 41.8 | 41.5 | 50.3 | 32.0 | 43.1 | 68.9 | 51.7 | 61.7 |
| **EvoProto (Resnet)** [Ours] | I | 70.8 | 63.6 | 68.2 | 75.4 | 53.6 | 70.9 | 41.3 | 43.2 | 42.4 | 51.7 | 37.6 | 46.4 | 69.8 | 56.1 | 64.0 |
| Joint (Resnet) | P | 78.4 | 76.4 | 77.4 | 79.8 | 70.2 | 77.4 | 47.8 | 46.9 | 47.7 | 78.4 | 76.4 | 77.4 | 78.4 | 76.4 | 77.4 |
| WILSON (ViT) Cermelli et al. (2022) | I | 74.3 | 61.2 | 68.8 | 75.2 | 46.4 | 69.1 | 41.0 | 42.3 | 41.8 | 52.6 | 36.1 | 46.2 | 73.6 | 57.5 | 66.9 |
| ToCo (ViT) Ru et al. (2023) | I | 73.5 | 58.8 | 67.4 | 74.6 | 44.3 | 67.9 | 40.3 | 41.4 | 41.1 | 49.2 | 33.9 | 43.4 | 72.9 | 56.0 | 65.7 |
| **EvoProto (ViT)** [Ours] | I | 75.9 | 66.7 | 72.3 | 78.2 | 52.6 | 72.8 | 42.9 | 44.5 | 43.9 | 58.7 | 43.0 | 52.5 | 74.7 | 64.5 | 70.7 |
| Joint(ViT) | P | 79.1 | 77.3 | 78.2 | 80.4 | 71.6 | 78.2 | 50.4 | 49.5 | 50.3 | 79.1 | 77.3 | 78.2 | 79.1 | 77.3 | 78.2 |

# 5 EXPERIMENTAL RESULTS

## 5.1 EXPERIMENTAL SETUP

**Datasets and Evaluation Metrics.** We evaluate EvoProto on Pascal VOC 2012 Everingham et al. (2010) and MS COCO Lin et al. (2014). Following the standard protocol Ahn & Kwak (2018b), the Pascal VOC training set is augmented to 10,582 images, and its validation set includes 1,449 images, both with 20 annotated classes. MS COCO is a large-scale dataset comprising 164K images with annotations spanning 80 object categories. For incremental evaluation, we report mIoU on the initial classes $C^0$, on the newly introduced classes $C^{1:T}$, and on all classes $C^{0:T}$.

**Protocols.** Following Cermelli et al. (2022); Si et al. (2024), we consider two incremental protocols: *disjoint*, where each step excludes future classes, and *overlap*, where new classes are introduced while retaining previously seen ones. Since the *overlap* scenario is more realistic and challenging, we focus on it, while results under the *disjoint* protocol are deferred to the Appendix A.1. Each scenario is denoted as $N_{\text{ini}}$-$N_{\text{inc}}$ (*e.g.*, 10-5 first trains on 10 classes and then adds 5 at each step). On VOC, we test four settings: 15-5 (2 steps), 10-10 (2 steps), 10-5 (3 steps), and 10-2 (6 steps). We also evaluate a challenging *COCO-to-VOC* setting, where 60 COCO classes are first learned, followed by incremental training on VOC, and finally evaluated on all 80 classes.

**Implementation Details.** We use ViT-B/16 Dosovitskiy et al. (2020) as the main backbone in all experiments to validate our method. The baseline is a plain ViT with an image-level classifier and a lightweight segmentation decoder from ToCo Ru et al. (2023). Both segmentation and classification tasks are supervised using the BCE loss. Following prior works Cermelli et al. (2022; 2020); Si et al. (2024), we also adopt Deeplab-V3 Chen (2017) with ResNet-101 He et al. (2016) on Pascal VOC and Wide-ResNet-38 Wu et al. (2019) on COCO-to-VOC to verify generalizability and ensure fair comparisons. The training setup follows ToCo Ru et al. (2023): a linear warm-up from $1 \times 10^{-6}$ to $6 \times 10^{-5}$ in the first 2k iterations, then polynomial decay (power 0.9). The initial step uses $6 \times 10^{-5}$ (20k iterations on VOC, 80k on COCO-to-VOC), while incremental steps use $2 \times 10^{-5}$ (8k on VOC, 20k on COCO-to-VOC). The batch size is set to 8. We apply *Confusion-Aware Prototype Evolving* after 4k iterations. Loss weights are $\lambda_{\text{cls}} = 1.0$, $\lambda_{\text{cl}} = 0.1$, $\lambda_{\text{kd}} = 0.1$, $\lambda_{\text{seg}} = 0.2$, with $k = 50$, $\gamma = 0.1$, $\tau = 0.1$.

Table 2: Ablation study of RCPL, CPA, and ALD on VOC 10–5, where "Initial (1–10)" is performance after training only on the first 10 classes (ViT-B/16).

| RCPL | CPA | ALD | Initial (1–10) | 1–10 | 11–20 | All |
|:---:|:---:|:---:|:---:|:---:|:---:|:---:|
|  |  |  | 80.1 | 72.9 | 56.0 | 65.7 |
| ✓ |  |  | 80.9 | 73.8 | 60.9 | 68.5 |
|  | ✓ | ✓ | 80.1 | 73.3 | 57.6 | 66.8 |
| ✓ | ✓ |  | 80.9 | 73.1 | 61.9 | 68.7 |
| ✓ |  | ✓ | 80.9 | 74.3 | 61.0 | 69.0 |
| ✓ | ✓ | ✓ | 80.9 | 74.7 | 64.5 | **70.7** |

## 5.2 QUANTITATIVE RESULTS

Leveraging the Confusion-Aware Prototype Evolving strategy and enhanced pseudo-label generation through Activation-Based Label Denoising, EvoProto significantly improves both model stability and plasticity. As shown in Tab. 1, it consistently outperforms prior approaches across all standard benchmarks, including recent methods Si et al. (2024) that benefit from SAM-based label refinement Kirillov et al. (2023a). Furthermore, the competitive results of our method with ViT-B/16 as the backbone demonstrating its strong capacity to generalize across different network architectures.

**10-10 VOC and 15-5 VOC.** Under the same ResNet-101 backbone, in the 10-10 VOC setting, we incrementally introduce 10 classes—dining table, dog, horse, motorbike, person, plant, sheep, sofa, train, and TV monitor. EvoProto achieves state-of-the-art performance on both old and new categories, outperforming the strongest prior methods (e.g., Teddy Si et al. (2024), Wilson Cermelli et al. (2022)) by at least 1.7% in overall mIoU. In 15-5 VOC, 5 additional VOC classes are introduced in the incremental step. Despite slightly lower performance on old classes and overall mIoU compared to Teddy Si et al. (2024), EvoProto achieves a 2.2% improvement in new classes mIoU, highlighting its superior capability in adapting to novel categories. The gains are primarily attributed to our strategies, and mitigation of class overwriting.

**10-2 VOC and 10-5 VOC.** These settings involve multiple incremental steps, intensifying catastrophic forgetting and class overwriting. Nevertheless, EvoProto achieves impressive performance. In the challenging 10-2 scenario, EvoProto surpasses Wilson Cermelli et al. (2022) and Teddy Si et al. (2024) by 13.9% and 3.3% in overall mIoU, respectively, and notably improves new-class performance by 14.2% and 5.6%. These results highlight EvoProto's superior capability in handling long-term incremental learning, a critical yet challenging practical scenario. Additionally, in the 10-5 task, EvoProto outperforms Wilson and Teddy by 5.9% and 2.3%, respectively.

**COCO-to-VOC.** In this more challenging setup, the model is initially trained on 60 COCO classes (none overlapping with VOC), followed by an incremental phase for 20 VOC classes with only image-level labels. The final evaluation spans all 80 classes in the COCO dataset. Tab. 1 shows that EvoProto maintains strong performance on both old and new classes, underscoring the effectiveness of our modules in balancing knowledge retention and new class learning. Despite the increased complexity and class diversity, EvoProto achieves state-of-the-art results in this setting as well.

## 6 ABLATION STUDY

**Component Ablations.** We analyze the main components of EvoProto—*Reweighted Contrastive Prototype Learning (RCPL)*, *Confusion-Aware Prototype Alignment (CPA)*, and *Activation-Based Label Denoising (ALD)*—on the VOC 10-5 task, which provides a balanced test of stability and plasticity. The baseline uses CAM pseudo-label supervision without additional enhancements, and results of different component combinations are shown in Tab. 2. Further ablations, including module design variants and experiments with ResNet101, are provided in the Appendix A.2.

Under identical settings, RCPL improves mIoU by 2.8%, showing that contrastive prototype learning reduces feature confusion and mitigates overwriting. Adding ALD yields gains of 1.6% on old and 2.6% on new classes by refining pseudo-labels, while CPA brings a further 3.5% improvement on new classes through confusion-aware distillation. Overall, EvoProto outperforms the baseline by 5.0% mIoU, confirming the combined effectiveness of its modules in alleviating semantic confusion.

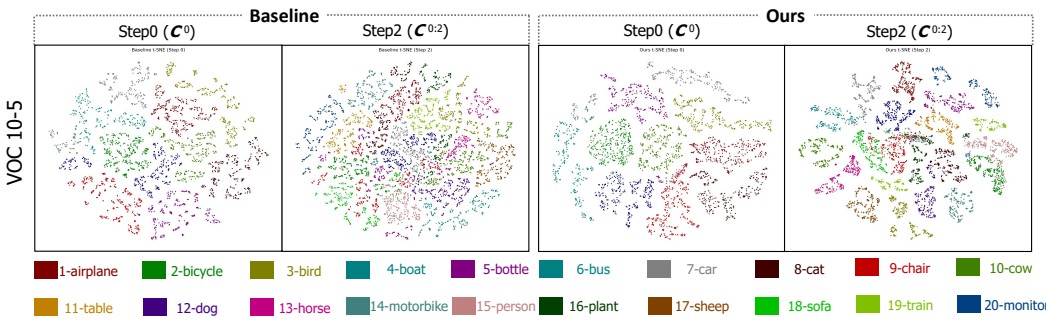

Figure 3: t-SNE visualization of internal feature distributions at initial and final incremental steps on the VOC 10-5 task. Different colors indicate different classes (excluding the background class for clarity). EvoProto maintains better inter-class separation compared to the baseline.

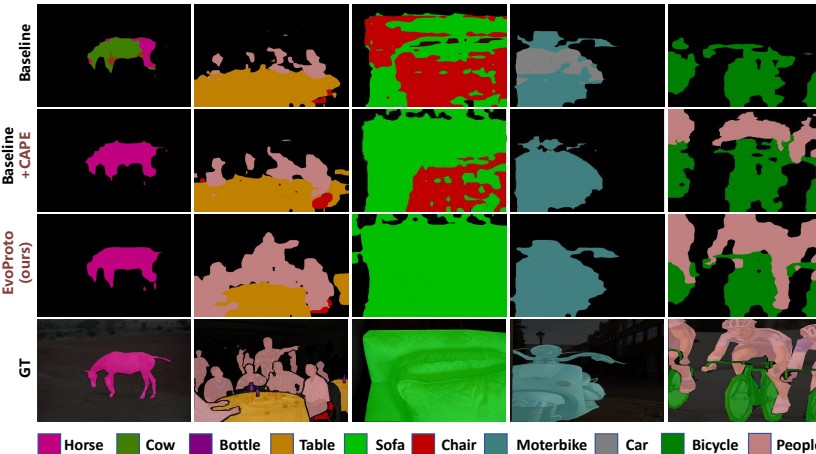

Figure 4: Qualitative segmentation results on the VOC 10-5 task. EvoProto significantly mitigates class overwriting and yields more accurate boundaries compared to *Baseline* and *Baseline+CAPE*.

**Interpretability Analysis.** We use t-SNE visualization (Fig. 3) to examine how EvoProto improves feature distributions during incremental learning. Under the VOC 10-5 setting, we compare baseline and EvoProto at the initial step ($C^0$) and final step ($C^{0:2}$). The baseline shows severe inter-class entanglement, while EvoProto preserves clear boundaries, highlighting its robustness in maintaining semantic distinctions and mitigating class overwriting.

**Qualitative Analysis.** We qualitatively assess VOC 10-5 results in Fig. 4, comparing the baseline, Baseline+CAPE, and EvoProto. In the first column, the baseline confuses similar categories (e.g., cow vs. horse), while our method resolves such errors. In the third and fourth columns, sofa is misclassified as chair and motorbike as car by the baseline, yet EvoProto correctly distinguishes them. Moreover, EvoProto produces sharper segmentation boundaries (second column). These results further demonstrate its effectiveness in alleviating class overwriting and improving WILSS performance. Additional qualitative examples are provided in the Appendix B.

## 7 CONCLUSION

In this paper, we identify class similarity as a key factor behind class overwriting in WILSS. To tackle this, we propose EvoProto, a novel WILSS framework that explicitly addresses class overwriting by evolving trainable and class-discriminative prototypes under weak supervision. Leveraging CAPE strategy and ALD mechanism, EvoProto alleviates inter-class confusion and improves pseudo-label quality. Extensive experiments demonstrate that our method preserves old knowledge while enabling effective adaptation to new classes, significantly mitigating class overwriting and confusion among similar categories.

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
