# EvoProto: Evolving Prototypes with Class Similarity for Weakly Supervised Incremental Segmentation

## Supplementary Material

## Contents

## A  ADDITIONAL EXPERIMENTS AND TRAINING DETAILS

In this section, we provide additional experimental results and training details as a supplement to the main paper.

### A.1  EXPERIMENTS ON DISJOINT SETTING

We also evaluate our method under the *disjoint* setting, where each step excludes future classes entirely, unlike the *overlap* setting where previously seen classes are retained alongside new ones. Tab. 1 reports the results, which shows that our method consistently outperforms other methods under this setting as well.

### A.2  ADDITIONAL ABLATION STUDY

**Component Ablation with ResNet-101 Backbone.** To further verify the generality of EvoProto's components across different backbone architectures, we repeat the component ablation study on the VOC 10-5 incremental task using ResNet101 as the backbone. The evaluation protocol and component combinations are kept identical to those in the main paper. As reported in Tab. 2 , the results are consistent with those under the ViT-B/16 backbone: each component individually contributes to performance improvement, while their combination yields the best balance between stability and plasticity. This demonstrates that the effectiveness of RCPL, CPA, and ALD is not tied to a specific backbone.

Table 1: Results on different disjoint settings. "P" and "I" denote pixel-level and image-level labels, respectively. Best image-level methods are in **bold**, best pixel-level methods are underlined. FT is a fine-tuning baseline (lower bound), Joint is trained with all classes (upper bound). Entries with "(ViT)" use a Vision Transformer backbone; others use ResNet.

| Method | Sup | 10-10 VOC | | | 15-5 VOC | | |
|--------|-----|------|-------|-----|------|-------|-----|
| | | 1-10 | 11-20 | All | 1-15 | 16-20 | All |
| FT | P | 7.7 | 60.8 | 33.0 | 8.4 | 33.5 | 14.4 |
| LWF [PAMI17] | P | 63.1 | 61.1 | 62.2 | 39.7 | 33.3 | 38.2 |
| LWF-MC [CVPR17] | P | 52.4 | 42.5 | 47.7 | 41.5 | 25.4 | 37.6 |
| ILT [ICCV19] | P | 67.7 | 61.3 | 64.7 | 31.5 | 25.1 | 30.0 |
| CIL [ITSC20] | P | 37.4 | 60.6 | 48.8 | 42.6 | 35.0 | 40.8 |
| MiB [CVPR20] | P | 66.9 | 57.5 | 62.4 | 71.8 | 43.3 | 64.7 |
| PLOP [CVPR21] | P | 63.7 | 60.2 | 63.4 | 71.0 | 42.8 | 64.3 |
| SDR [CVPR21] | P | 67.5 | 57.9 | 62.9 | 73.5 | 47.3 | 67.2 |
| RECALL [ICCV21] | P | 64.1 | 56.9 | 61.9 | 69.2 | 52.9 | 66.3 |
| CAM [CVPR16] | I | 65.3 | 41.3 | 54.5 | 69.3 | 26.1 | 59.4 |
| SEAM [CVPR20] | I | 65.1 | 53.5 | 60.6 | 71.0 | 33.1 | 62.7 |
| SS [CVPR20] | I | 60.7 | 25.7 | 45.0 | 71.6 | 26.0 | 61.5 |
| EPS [CVPR21] | I | 64.2 | 54.1 | 60.6 | 72.4 | 28.5 | 65.2 |
| WILSON [CVPR22] | I | 64.5 | 54.3 | 60.8 | 73.6 | 43.8 | 67.3 |
| Teddy [ECCV24] | I | 65.4 | 55.2 | 61.7 | 74.5 | 48.1 | 69.0 |
| **EvoProto(Resnet)** [Ours] | I | **66.7** | **57.8** | **63.4** | **75.0** | **51.1** | **70.1** |
| Joint(Resnet) | P | 78.4 | 76.4 | 77.4 | 79.8 | 72.4 | 77.4 |
| WILSON (ViT) [CVPR22] | I | 67.9 | 58.4 | 64.5 | 74.8 | 44.5 | 68.2 |
| ToCo (ViT) [CVPR23] | I | 66.3 | 57.2 | 63.2 | 74.1 | 43.3 | 67.4 |
| **EvoProto(ViT)** [Ours] | I | **71.5** | **60.2** | **67.1** | **77.6** | **53.0** | **72.5** |
| Joint(ViT) | P | 79.1 | 77.3 | 78.2 | 80.4 | 76.0 | 78.2 |

Table 2: Ablation study of RCPL, CPA, and ALD on the VOC 10-5 task (ResNet-101).

| RCPL | CPA | ALD | 10-5 (3 steps) | | |
|------|-----|-----|------|-------|-----|
| | | | 1–10 | 11–20 | All |
| | | | 67.9 | 45.8 | 58.4 |
| ✓ | | | 69.0 | 52.6 | 62.1 |
| | ✓ | ✓ | 68.8 | 50.0 | 60.8 |
| ✓ | ✓ | | 69.3 | 53.0 | 62.4 |
| ✓ | | ✓ | 69.7 | 53.5 | 62.9 |
| ✓ | ✓ | ✓ | 69.8 | 56.1 | **64.0** |

**Ablation Study of ALD.** To further investigate the effectiveness of the ALD module, we conduct ablation studies under the VOC 10-2 setting. Specifically, we evaluate the F1 classification score over incremental steps, the CAM mIoU across all classes, and the CAM mIoU of confusing categories such as cow and sheep. These metrics are compared between models trained with and without the ALD module, as illustrated in Fig. 1. The results demonstrate that ALD can effectively filter out noisy CAM pseudo-labels and provide higher-quality supervision signals. Thereby alleviating the class overwriting problem, which is exacerbated in weakly supervised settings due to the absence of pixel-level annotations.

## A.3 ADDITIONAL ABLATION ON THE CONFUSION SCORE

In the main paper, we have explained the motivation and formulation of the proposed confusion score. To rigorously validate our conclusion—that the confusion score effectively captures the degree of confusion between new and old classes, with higher values indicating stronger confusion—we conduct additional ablation experiments.

As previously discussed, the similarity between class representations in the encoder's feature space reflects semantic affinity and potential confusion. Leveraging this insight, we design an experiment to assess the ability of our confusion score to approximate inter-class confusion in the absence of pixel-level annotations.

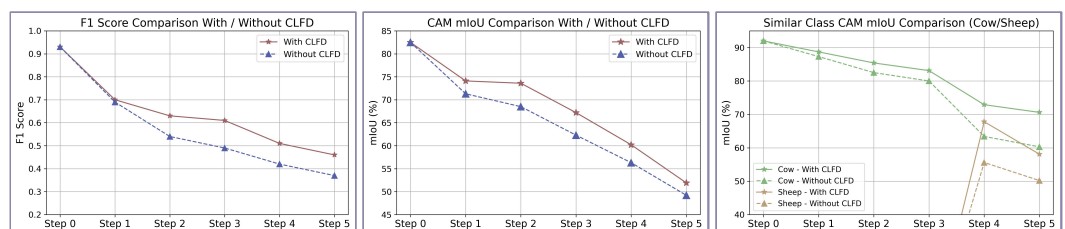

Figure 1: Ablation of the ALD module on VOC 10-2 task. Comparison of F1 score, overall mIoU, and mIoU of confusing classes (e.g., cow and sheep) with and without ALD.

Table 3: Most similar old class for each new class in the 10-5 setting, identified by GT-trained and EvoPrototype models.

| Method | diningtable | dog | horse | motorbike | person | pottedplant | sheep | sofa | train | tvmonitor |
|---|---|---|---|---|---|---|---|---|---|---|
| GT-trained | chair | cat | cow | bike | bus | chair | cow | chair | bus | person |
| EvoPrototype | chair | cat | cow | bike | bus | diningtable* | cow | chair | bus | chair* |

Specifically, we compare the most confusing class pairs identified by our confusion score with those derived from a model trained using ground-truth dense annotations. We evaluate both the GT-trained model and our EvoPrototype model on the 10-5 incremental setting. For the GT-trained model, we extract class-level representations using EMA-aggregated features, while for EvoPrototype, we use the learnable prototypes. In both cases, cosine similarity is used to quantify the inter-class affinity.

As shown in Tab. 3, our confusion score aligns well with feature-based similarity, demonstrating its effectiveness in identifying semantically entangled class pairs even without dense supervision.

### A.4 HYPERPARAMETER ANALYSIS

We further analyze the sensitivity of EvoProto to the two hyperparameters $\tau$ and $\delta$, which control the scaling of the weight matrix and the separation between class prototypes, respectively. Table 4 summarizes the results under both the disjoint and overlap protocols on the VOC 10–5 setting.

We observe that EvoProto is generally robust across a wide range of values. For $\gamma$, performance remains stable when varying from 0.01 to 0.2, with the best overall accuracy achieved at $\gamma = 0.1$. Similarly, when fixing $\gamma = 0.1$ and varying $\tau$, results remain consistent, with $\tau = 0.1$ yielding the strongest performance in both protocols. These results confirm that EvoProto is not overly sensitive to the choice of hyperparameters, while values ($\tau = 0.1$, $\gamma = 0.1$) provide the best balance.

### A.5 CONFUSION SCORE NORMALIZATION FUNCTION.

Since the raw confusion scores are relatively small (ranging from $10^{-4}$ to $10^{-2}$), we explore various normalization strategies to map them into a suitable range for downstream weighting purposes, as summarized in Tab. 5.

We first consider min-max normalization, which effectively highlights the relative confusion relationships among different classes within the same incremental step. However, directly mapping the minimum score to $-1$ and the maximum to $1$ can lead to disproportionate fluctuations and offsets—especially when the actual confusion levels among classes are close—thereby amplifying minor differences and introducing instability in representation learning.

Next, we explore a mean-centered linear normalization strategy, which uses the mean confusion score as the normalization anchor. This approach addresses the drawbacks of min-max normalization by avoiding extreme scaling when class-wise confusion values are similar, thereby reducing instability. However, compared to nonlinear alternatives such as the sigmoid-based normalization, the linear method lacks the ability to smoothly compress outliers and emphasize moderate variations.

Table 4: Hyperparameter Analysis of EvoProto on $\tau$ and $\gamma$ under the 10-5 VOC setting.

| $\tau$ | $\gamma$ | Disjoint | | | Overlap | | |
|---|---|---|---|---|---|---|---|
| | | 1–10 | 11–20 | All | 1–10 | 11–20 | All |
| | 0.01 | 69.3 | 58.3 | 65.1 | 74.0 | 63.6 | 70.0 |
| | 0.05 | 69.3 | 58.7 | 65.3 | 74.2 | 65.0 | 70.3 |
| **0.1** | **0.1** | 69.8 | 59.5 | **66.0** | 74.7 | 64.5 | **70.7** |
| | 0.15 | 69.5 | 58.1 | 65.1 | 73.9 | 63.3 | 69.8 |
| | 0.2 | 69.1 | 57.5 | 64.6 | 73.3 | 62.5 | 69.1 |
| 0.025 | | 69.5 | 57.6 | 64.9 | 73.7 | 62.5 | 69.3 |
| 0.05 | | 69.6 | 58.8 | 65.5 | 73.8 | 62.8 | 69.5 |
| **0.1** | **0.1** | 69.8 | 59.5 | **66.0** | 74.7 | 64.5 | **70.7** |
| 0.15 | | 69.3 | 59.6 | 65.8 | 74.2 | 64.0 | 70.2 |
| 0.2 | | 68.9 | 58.4 | 65.0 | 73.2 | 62.8 | 69.1 |

As a result, it may underutilize the semantic distinction between highly confused and less confused class pairs, leading to suboptimal modulation in weighting-sensitive components like distillation and contrastive learning.

Finally, we adopt a **centered sigmoid normalization** strategy. Compared with both min-max and mean-centered linear normalization, the centered sigmoid method achieves notably superior performance across both old and new classes. This demonstrates that the smooth and bounded characteristics of the sigmoid function, when centered around the mean confusion score, are better suited for generating adaptive weights that effectively modulate distillation strength and feature separation during the incremental learning process.

Table 5: Segmentation performance (%) of different normalization methods on new classes, old classes, and overall.

| Normalization Method | New Classes | Old Classes | Overall |
|---|---|---|---|
| None | 63.5 | 69.5 | 67.1 |
| Min-Max Norm | 64.6 | 71.9 | 68.8 |
| Mean-Centered Linear Norm | 65.1 | 72.0 | 69.2 |
| Centered Sigmoid Norm | 64.5 | 74.7 | 70.7 |

## A.6 EVOPROTOTYPE VS. SAMPLING-BASED PROTOTYPE AVERAGING

Our method introduces a set of evolving prototypes that are dynamically updated throughout the training process. In contrast, a common alternative is to compute prototypes by directly averaging the features of masked regions (i.e., sampling-based averaging). However, this naive averaging approach suffers from two main drawbacks: (1) it treats all pixels within the class mask equally, regardless of their semantic relevance or confidence, and (2) it is sensitive to noise in pseudo labels, especially under weak supervision, which can lead to biased or ambiguous prototypes.

In comparison, EvoPrototype employs learnable parameters that are continuously updated along with the network, thereby alleviating the influence of incomplete supervision and noisy labels. Moreover, since EvoPrototype avoids direct averaging operations, it better preserves intra-class feature diversity, which is crucial for improving the effectiveness of prototype-based distillation and class separation.

To validate the effectiveness of our evolving strategy, we conduct ablation experiments comparing EvoPrototype with the conventional sampling-based method, as shown in Tab. 6. EvoPrototype consistently outperforms both the sampling-based strategy and the baseline without prototype guidance across all evaluation subsets. Specifically, EvoPrototype achieves an absolute improvement of **5.2%** on new classes (74.7 vs. 69.5) and **3.8%** on confusing classes (69.9 vs. 66.1) compared to the sampling-based method. For old classes, EvoPrototype also yields a moderate gain of **0.3%** (64.5 vs. 64.2). Overall, EvoPrototype brings the average performance across all classes from 64.0% to **70.7%**, demonstrating its superiority in maintaining class discrimination and robustness to pseudo-label noise in both novel and ambiguous categories.

Table 6: Ablation study comparing different prototype strategies (None, Sampling-based, and Evo-Prototype) across various class types. EvoPrototype demonstrates superior performance by dynamically learning robust class representations, effectively mitigating the limitations of naive averaging under weak supervision.

| Prototype Strategy | New Classes | Old Classes | Confusing Classes | All Classes |
|---|---|---|---|---|
| None | 66.0 | 62.1 | 62.8 | 64.0 |
| Sampling-based | 69.5 | 64.2 | 66.1 | 66.8 |
| EvoPrototype | **74.7** | **64.5** | **69.9** | **70.7** |

Table 7: Prototype cosine similarity and average segmentation mIoU (%) for three visually similar class pairs under different supervision regimes. Weak supervision increases prototype similarity and degrades segmentation accuracy due to semantic confusion. Our EvoPrototype reduces prototype similarity and improves segmentation performance, mitigating class overwriting.

| Class Pair | Metric | Fully Sup. | Weakly Sup. | W.S. + EvoProto |
|---|---|---|---|---|
| (cow, sheep) | Cosine Sim. ↓ | 0.34 | 0.59 | **0.18** |
| | mIoU (%) ↑ | 80.1 | 65.4 | **75.3** |
| (train, bus) | Cosine Sim. ↓ | 0.28 | 0.63 | **0.21** |
| | mIoU (%) ↑ | 68.9 | 45.6 | **60.3** |
| (sofa, chair) | Cosine Sim. ↓ | 0.18 | 0.57 | **0.23** |
| | mIoU (%) ↑ | 48.1 | 29.1 | **40.5** |

A.7   ANALYSIS OF CLASS OVERWRITING CAUSED BY VISUALLY SIMILAR CATEGORIES

As discussed in the main text, visually similar categories are more prone to class overwriting in the context of weakly supervised incremental learning, where supervision is often incomplete. To verify this hypothesis, we conduct an oracle experiment to quantitatively assess both the feature-level similarity of such class pairs and their impact on the final segmentation results.

We employ ViT-B/16 as the baseline backbone and conduct experiments on the 10-5 incremental setting under three supervision regimes: fully supervised, weakly supervised, and weakly supervised with our proposed EvoPrototype optimization. To evaluate feature-level similarity, we select three representative pairs of visually similar classes—(cow, sheep), (train, bus), and (sofa, chair)—and compute the cosine similarity between their class prototypes. Simultaneously, we measure the average segmentation mIoU for these confusing class pairs.

As shown in Tab. 7, we draw two main conclusions: First, higher feature-level similarity between visually similar classes significantly degrades the model's segmentation performance on these pairs, confirming the impact of class confusion. Second, compared to full supervision, weak supervision leads to higher prototype similarity for confusing pairs, which correlates with a notable drop in segmentation accuracy. In contrast, our EvoPrototype—by continuously evolving prototypes and adaptively controlling the distillation and separation strengths—substantially reduces prototype similarity, better preserves class discriminability, and effectively mitigates the class overwriting phenomenon. On average, our method improves the mIoU for confusing pairs by over 10% compared to the weakly supervised baseline.

A.8   QUANTITATIVE ANALYSIS FOR CLASS OVERWRITING

While the qualitative illustration in Fig. 1 of the main paper provides an intuitive understanding of the *class overwriting* issue, we agree that a quantitative analysis strengthens the motivation. We therefore add a controlled study on VOC (in Tab. 8 below).

*Protocol.* Train on 15 classes, then introduce a semantically similar new class (e.g., *bus→train*, *chair→sofa*, *cow→sheep*). We report the old-class mIoU before the step ("Old (Init)"), after incremental training with the baseline ("Old (Base)") and with ours ("Old (Ours)"), together with the new-

class mIoU ("New (Base/Ours)"). The baseline is a plain ViT with an image-level classifier and a lightweight decoder, supervised only by BCE for classification/segmentation (no RCPL/CPA/ALD).

*Findings (VOC).* As shown in Tab. 8, the baseline exhibits marked drops on old classes when a similar new class is added (e.g., *bus*: $-17.8\%$; *chair*: $-21.7\%$; *cow*: $-24.5\%$) and learns the new class poorly. EvoProto substantially mitigates these drops (to $-2.6\%$, $-10.5\%$, $-6.4\%$ respectively) while boosting the new-class mIoU by $+10\%$ to $+18\%$, quantitatively evidencing class overwriting and our mitigation.

Table 8: Impact of confusing new classes on both old and new class performance. Our method alleviates class overwriting and improves new class learning.

| Old Class | New Class | Old (Init) | Old (Base) | Old (Ours) | New (Base) | New (Ours) |
|---|---|---|---|---|---|---|
| Bus | Train | 90.1 | 72.3 ($\downarrow$17.8) | **87.5 ($\downarrow$2.6)** | 48.7 | **59.4** |
| Chair | Sofa | 53.3 | 31.6 ($\downarrow$21.7) | **42.8 ($\downarrow$10.5)** | 28.9 | **42.4** |
| Cow | Sheep | 91.7 | 67.2 ($\downarrow$24.5) | **85.3 ($\downarrow$6.4)** | 62.3 | **79.8** |

## A.9 Prototypes Training Procedure

As briefly mentioned in Sec. 4.1.2 of the main paper, due to space constraints, we only provided a simplified explanation that class prototypes are optimized using the binary cross-entropy (BCE) loss. In this section, we present a more comprehensive description of the prototype training process. Specifically, Alg. 1 details the full training procedure, including initialization, feature extraction, similarity computation, and loss-based optimization of class prototypes.

---

**Algorithm 1:** Class Prototype Training for Incremental Semantic Segmentation

---

**Input** : Image batch $x$, pseudo label $y$, task-wise class list $\{C^0, \ldots, C^t\}$, temperature $T$

**Output:** Trained prototypes $\{P^{(i)}\}$ for all learned classes $\bigcup_{i=0}^{t} C^i$

**Initialization:**
 **foreach** *task* $k \in \{0, \ldots, t-1\}$ **do**
  └ Load prototype matrix $\mathbf{P}_k$ from previous model

 Initialize prototype matrix for current task $t$: $\mathbf{P}_t \in \mathbb{R}^{|C^t| \times d}$

 Aggregate all prototypes: $\mathbf{P} \leftarrow \text{Concat}\left(\{\text{Normalize}(\mathbf{P}_k)\}_{k=0}^{t}\right)$

**Feature Extraction:**
 Compute pixel-level embedding feature map: $f \leftarrow \text{Backbone}(x)$
 Flatten and normalize features: $f_{\text{flat}} \leftarrow \text{Normalize}(\text{Flatten}(f)) \in \mathbb{R}^{(BHW) \times d}$

**Similarity Computation:**
 Compute similarity logits: $\text{sim\_logits} \leftarrow f_{\text{flat}} \cdot \mathbf{P}^\top$
 Reshape to segmentation map: $\hat{y}_{\text{sim}} \leftarrow \text{Reshape}(\text{sim\_logits}) \in \mathbb{R}^{B \times |C^{0:t}| \times H \times W}$
 Temperature scaling: $\hat{y}_{\text{sim}} \leftarrow \frac{\hat{y}_{\text{sim}}}{T}$

**Loss Computation:**
 Upsample predictions to target size
 Convert pseudo label $y$ to one-hot: $y_{\text{onehot}} \in \{0, 1\}^{B \times C \times H \times W}$
 Define ignore mask: $M \leftarrow (y \neq 255)$
 Apply BCE loss:
 $\mathcal{L}_{\text{p\_seg}} \leftarrow \frac{1}{\sum M} \sum M \cdot \mathcal{L}_{\text{BCE}}(\hat{y}_{\text{sim}}, y_{\text{onehot}})$

**Optimization:**
 Backpropagate and update prototypes with $\mathcal{L}_{\text{p\_seg}}$

---

# B ADDITIONAL QUALITATIVE RESULTS

In this section, we present further qualitative results to validate the effectiveness of our proposed method. We first compare EvoProto with baseline methods and intermediate variants to highlight improvements in segmentation accuracy and class discrimination in Sec. B.1. Then, we demonstrate how EvoProto performs across incremental learning steps, showcasing its robustness in adapting to new classes while retaining previously acquired knowledge in Sec. B.2. Finally, in Sec. B.3, we provide a qualitative comparison of Class Activation Maps (CAMs) generated with and without the ALD module, illustrating its impact on filtering out unreliable activations to generate higher-quality pseudo labels.

## B.1 QUALITATIVE COMPARISON WITH BASELINE AND PARTIAL MODELS

In this section, we provide additional qualitative analysis by comparing our proposed EvoProto method (3rd column) with both the Baseline (1st column) and Baseline+CAPE (2nd column) under the 10-5 scenario in CISS, as illustrated in Fig. 2 and Fig. 3.

In Fig. 2, the 1st and 3rd rows show that the Baseline method produces very blurred segmentation boundaries for the *airplane* and *horse* classes, indicating poor spatial precision. In contrast, with the addition of the CAPE module and especially with EvoProto, the predicted boundaries become much more aligned with the ground truth, significantly improving segmentation quality. This reflects EvoProto's ability to capture fine-grained details and maintain sharper object contours. In the 2nd and 5th rows, due to the class overwriting phenomenon, the Baseline method misclassifies visually similar categories—such as *horses* and *cows*—as the newly learned class, *sheep*. Our Evo-Proto method, on the other hand, effectively distinguishes among these semantically similar classes and produces accurate segmentation, demonstrating robustness in resolving semantic ambiguity and inter-class confusion. Furthermore, in the 4th row, EvoProto clearly outperforms both Baseline and Baseline+CAPE in separating *sofa* from *chair*, which often exhibit close visual resemblance in complex scenes. In the 7th row, while the Baseline and Baseline+CAPE methods are only capable of segmenting the old class *bicycle* and fail to identify the newly introduced class *person*, Evo-Proto successfully segments both objects. This illustrates EvoProto's balanced ability to preserve knowledge of old classes while effectively adapting to new ones.

Additionally, EvoProto consistently achieves more refined and coherent segmentation boundaries across different categories, highlighting its superior boundary-level understanding. Fig. 3 further presents additional examples that emphasize EvoProto's strengths in distinguishing between visually or semantically similar old and new classes. It also demonstrates EvoProto's effectiveness in alleviating the class overwriting problem and maintaining consistency across incremental steps. These comprehensive qualitative results further validate that EvoProto not only enhances spatial precision but also significantly mitigates the conflict between old class retention and new class acquisition, offering a reliable solution for continual semantic segmentation tasks.

## B.2 QUALITATIVE RESULTS ACROSS INCREMENTAL STEPS

In this section, we present qualitative results that illustrate the performance of our proposed Evo-Proto method across incremental learning steps, as shown in Fig. 4. Specifically, we visualize the segmentation outcomes on the same image under the 10-2 scenario as the model progressively learns new classes.

In the 1st column of Fig. 4, before the model encounters the class *train*, it misclassifies it as the previously learned class *bus* due to their semantic similarity. After learning *train* at step 5, the model successfully distinguishes between these two visually and semantically similar classes. This demonstrates EvoProto's effectiveness in reducing class confusion and mitigating misclassification caused by the class overwriting phenomenon. Similarly, in the 2nd and 4th columns, prior to learning the classes *horse* and *sheep*, the model erroneously segments them as the old class *cow*. Once these new classes are introduced, EvoProto is able to accurately segment them, showcasing its ability to update its knowledge without forgetting previously learned semantics.

Furthermore, the 3rd and 6th columns demonstrate that even after learning semantically similar new classes—*sheep* (similar to *cow*) and *sofa* (similar to *chair*)—the model does not confuse them with

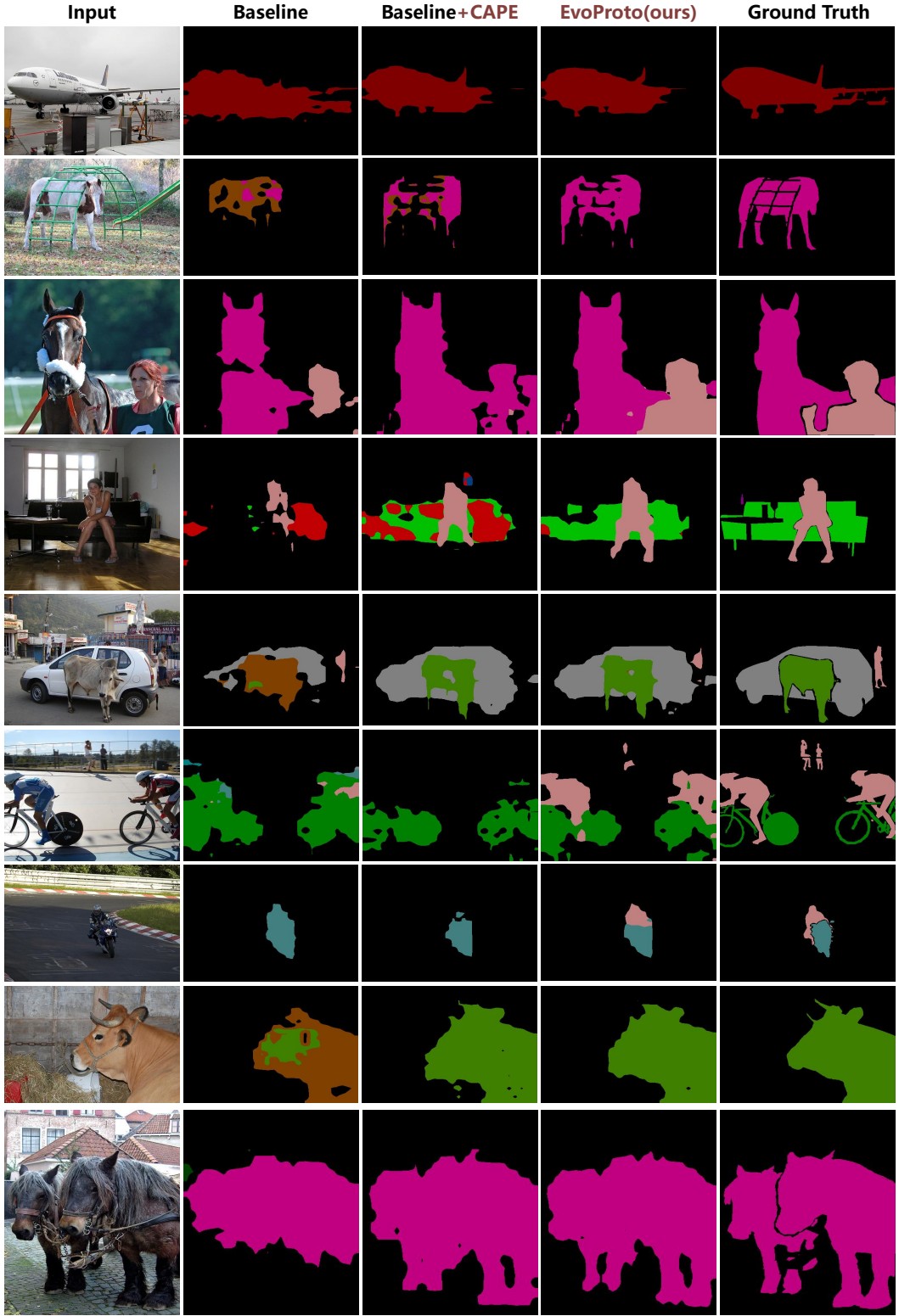

| Input | Baseline | Baseline+CAPE | EvoProto(ours) | Ground Truth |

Figure 2: Qualitative results of EvoProto comparing to Baseline, Baseline+CAPE on 10-5 task of VOC. Each class is uniquely represented by a specific color, making both boundary accuracy and correct color alignment with the ground truth essential for evaluation.

| Input | Baseline | Baseline+CAPE | EvoProto(ours) | Ground Truth |
|---|---|---|---|---|

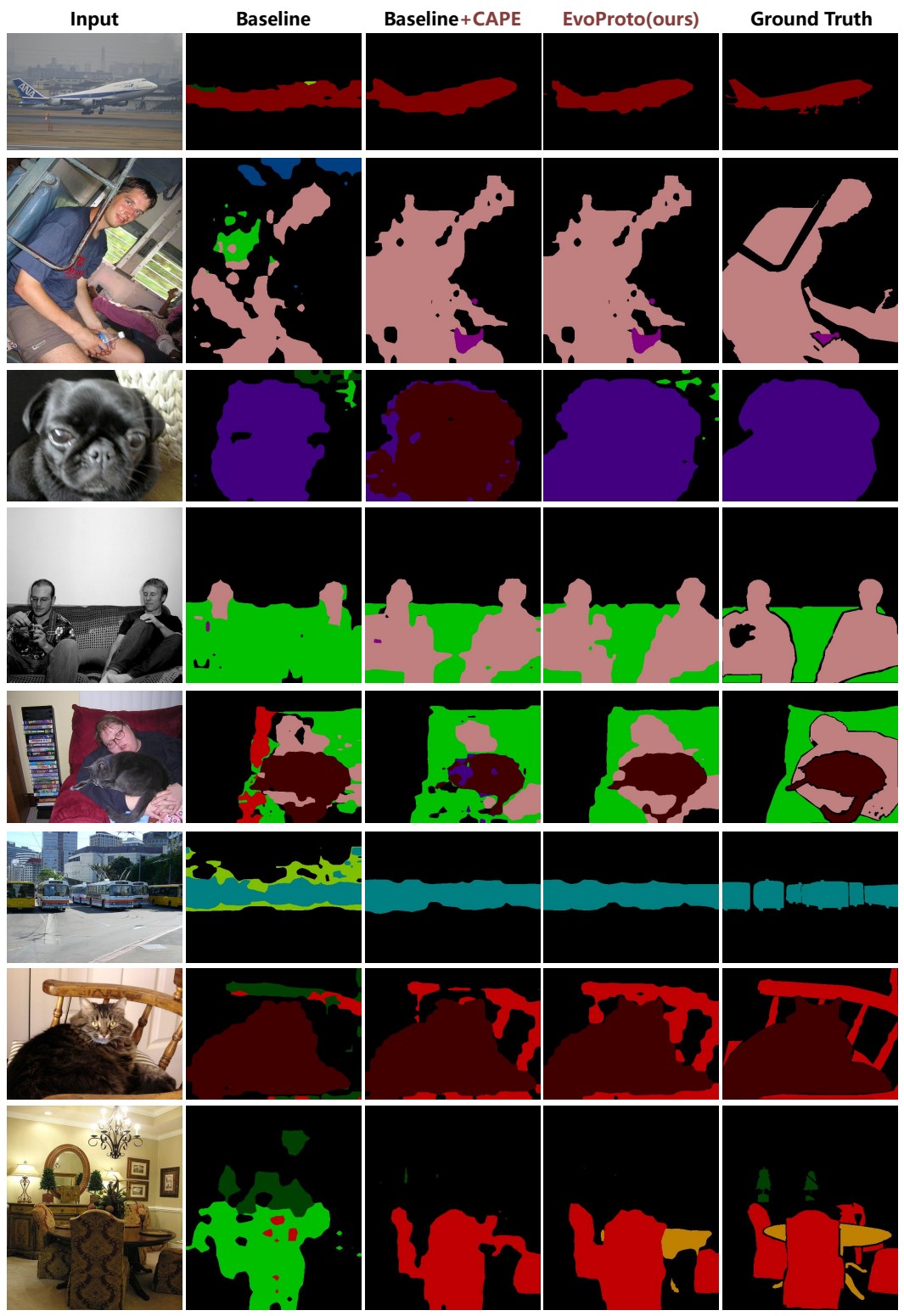

Figure 3: Qualitative results of EvoProto comparing to Baseline, Baseline+CAPE on 10-5 task of VOC. Each class is uniquely represented by a specific color, making both boundary accuracy and correct color alignment with the ground truth essential for evaluation.

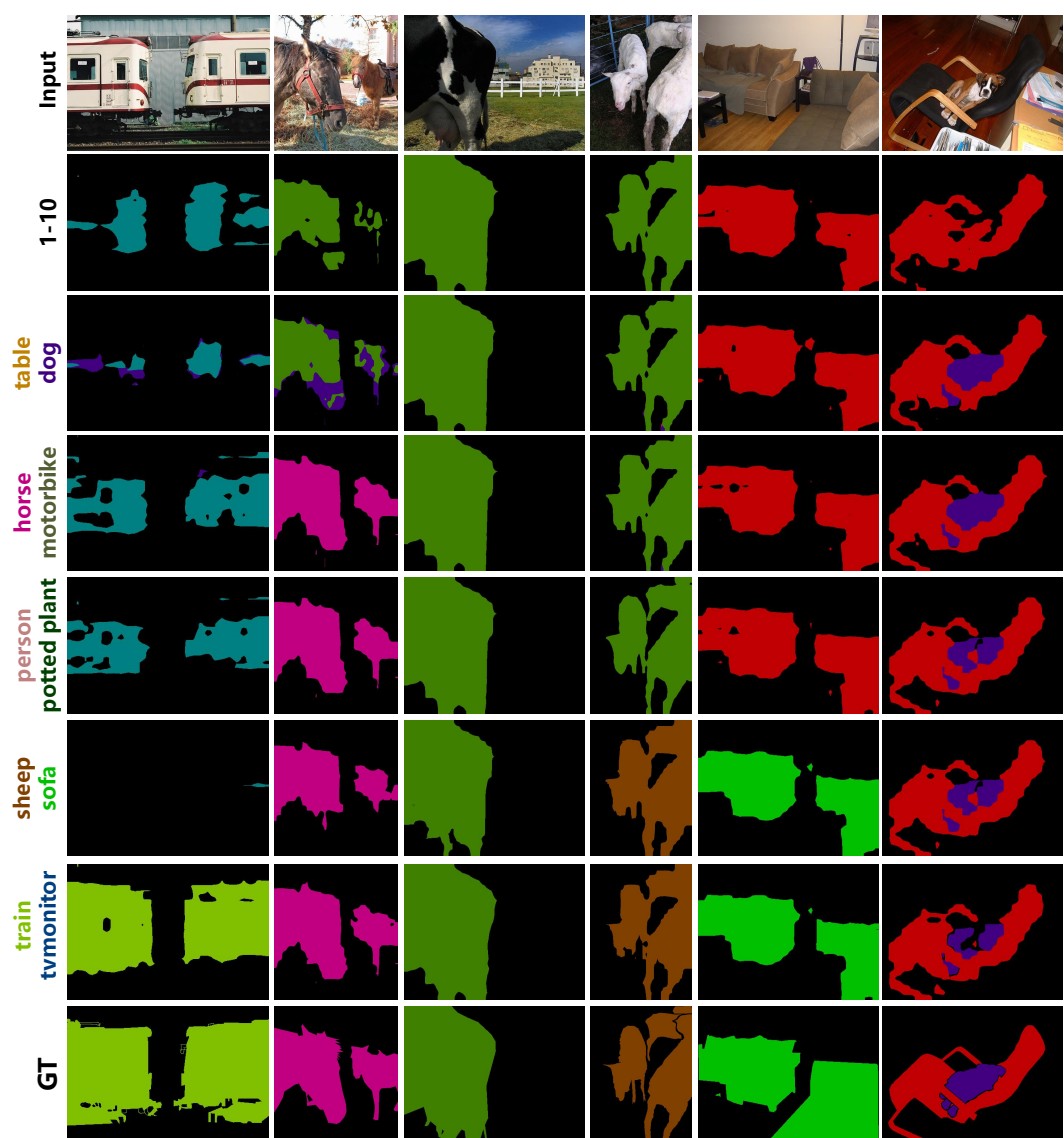

Figure 4: Qualitative results of EvoProto across incremental steps on the 10-2 task of VOC, the left-most region indicates the novel classes introduced at each incremental step. Each class is uniquely represented by a specific color, making both boundary accuracy and correct color alignment with the ground truth essential for evaluation.

the old ones. This indicates that EvoProto preserves clear class boundaries and maintains consistency between old and new knowledge, thereby effectively mitigating class overwriting.

Overall, this visualization highlights EvoProto's ability to continuously integrate new information while preserving the segmentation accuracy of previously learned classes, confirming its robustness in continual semantic segmentation.

### B.3 QUALITATIVE COMPARISON OF CAMS WITH AND WITHOUT ALD MODULE

In this section, we present the qualitative comparison of CAMs generated with and without the ALD module to illustrate its effectiveness, as shown in Fig. 5. Each row in the figure consists of four columns: the input image, the CAM produced by our method without the ALD module, the CAM with the ALD module, and the ground truth. It is evident that incorporating the ALD module

| Input | EvoProto - ALD | EvoProto | Ground Truth |
|---|---|---|---|

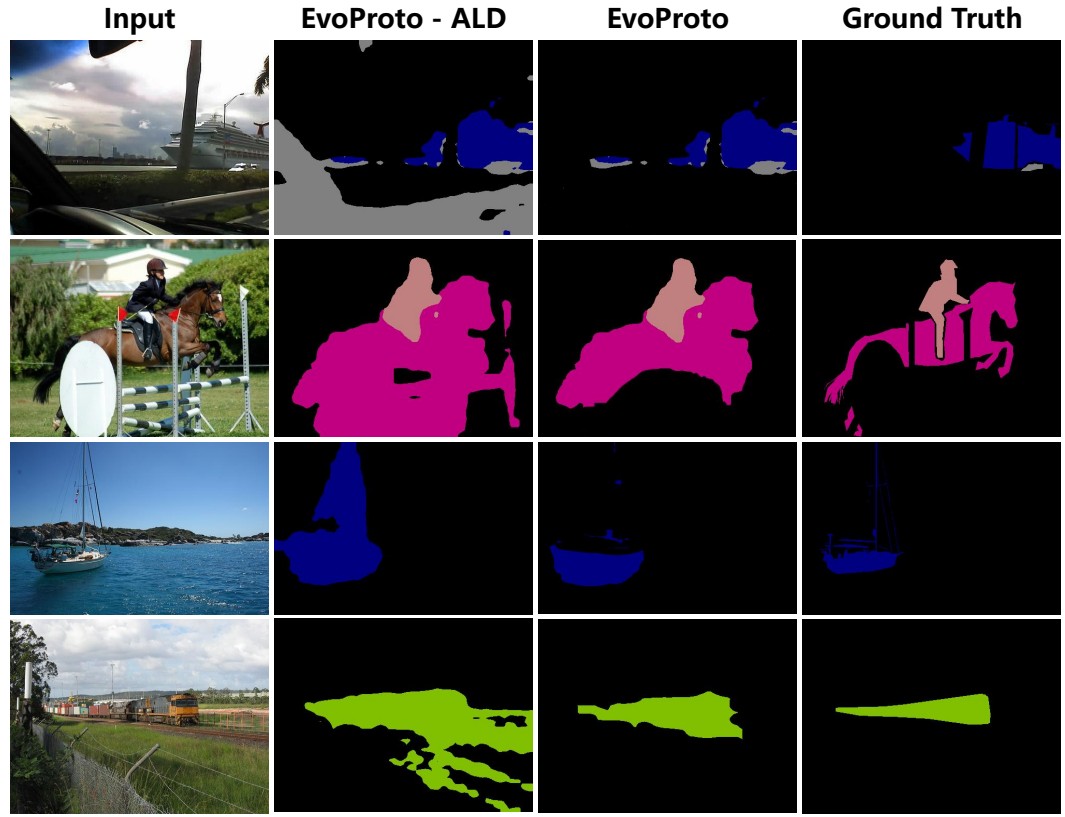

Figure 5: Qualitative comparison of CAMs with and without the ALD module on the 10-5 VOC task. Each class is uniquely represented by a specific color, making both boundary accuracy and correct color alignment with the ground truth essential for evaluation.

effectively suppresses noise in the CAMs, resulting in pseudo-labels with boundaries more closely aligned to the ground truth. Such high-quality pseudo-labels are crucial in WILSS task, where pixel-level annotations are unavailable during the incremental steps. By providing cleaner supervision signals, the ALD-enhanced CAMs facilitate better learning in each step and help mitigate the class overwriting problem, thereby demonstrating the practical value of the ALD module.

## C  LIMITATION AND BROADER IMPACT

From the visualization results, we observe that our method still encounters challenges in accurately delineating object boundaries, which suggests that there is considerable room for improvement in modeling fine-grained structures and spatial details. Addressing this limitation may require incorporating boundary-aware modules or leveraging higher-resolution feature representations. Furthermore, our current empirical evaluation has been restricted to the Pascal VOC and COCO benchmarks. While these datasets are widely adopted, they are relatively less complex compared to large-scale benchmarks such as ADE20K. Extending our evaluation to ADE20K and other challenging datasets will provide a more comprehensive understanding of the generalization ability of our approach. Moreover, although our study primarily focuses on the weakly-incremental learning setting (WILSS), we emphasize that the issue of class overwriting is not unique to this scenario; it is also prevalent in fully supervised incremental segmentation. Therefore, the proposed strategies have the potential to contribute to a broader spectrum of incremental segmentation tasks beyond the weakly supervised setting.