# OpenReview forum: "EvoProto: Evolving Prototypes with Class Similarity for Weakly Incremental Segmentation"
_ICLR.cc/2026/Conference — Submitted to ICLR 2026_

### Official Review · Reviewer_og6J · 2025-10-30

**Soundness:** 3
**Presentation:** 3
**Contribution:** 2
**Rating:** 6
**Confidence:** 5

**Summary:**

This paper addresses the problem of Weakly Supervised Incremental Learning for Semantic Segmentation (WILSS), where models must learn to segment new classes with only image-level labels, without accessing previous class data. The authors identify class overwriting， the misclassification of old classes as new ones due to class confusion, as a key failure mode in WILSS. This paper provide a framework that explicitly models and mitigates class confusion through dynamic prototype evolution guided by inter-class similarity. Experiments on Pascal VOC  and COCO-to-VOC under multiple incremental protocols (10-10, 15-5, 10-5, 10-2) show state-of-the-art results.

**Strengths:**

After reading this paper carefully, I think this paper has the following strengths:

- The idea of quantifying inter-class confusion and evolving prototypes accordingly is novel and well-motivated for the WILSS setting. Integrating contrastive learning and prototype-level distillation under confusion-aware reweighting is a creative and non-trivial combination.

- The paper is generally well-written and structured, with logical flow from motivation to implementation.

- The method is rigorously formulated, with mathematical clarity on confusion scoring, reweighting, and objective functions.

- The paper targets an emerging and challenging area (WILSS) with practical importance: learning segmentation models from limited supervision under continual updates.

**Weaknesses:**

The following weaknesses should be solved to improve the quality of this paper:

- While the confusion-aware weighting is new, the prototype evolution concept shares similarities with prior prototype refinement works such as RePRIand PLOP. The contribution could be perceived as incremental rather than fundamentally novel.

- The paper defines a symmetric confusion score from pseudo-labels but does not analyze its robustness to CAM noise or how it behaves across steps. It’s unclear whether confusion scores remain stable or amplify errors over time; this could undermine reliability.

- The framework introduces multiple prototype-level losses and dynamic confusion estimation per epoch. Runtime or memory overhead is not discussed.

- The ALD module uses activation thresholds but the rationale for the averaging bias “+1” and its hyperparameter sensitivity is underexplained. The section could be clearer on its interaction with confusion-aware objectives.

**Questions:**

I have some questions fior this paper:

Q1: How does the model prevent confusion scores from being corrupted by noisy pseudo-labels, especially early in incremental steps? Have the authors tried smoothing or memory-based averaging?

Q2: Would EvoProto generalize to instance segmentation or open-vocabulary segmentation tasks?

---

> ### Author Response · Authors · 2025-11-20
>
> We thank the reviewer for the constructive assessment and insightful comments.
> Below, we address all concerns point-by-point.
>
> **R4-W1: Contributions of *EvoProto***
>
> We would like to further clarify what we believe are the key contributions of *EvoProto*.
>
> First, we appreciate your recognition of our confusion-aware mechanism.
> This mechanism equips the network with the ability to explicitly identify and handle confused class pairs that naturally emerge during incremental learning.
> By doing so, it enhances both the plasticity and stability of the model, particularly for categories where previous works often struggle.
> We view confusion awareness as a novel and effective remedy for the widely observed class overwriting problem in WILSS, and the analyses and experiments in Supplementary A.7 strongly support this claim.
>
> Second, although both our method and prior works rely on prototypes to improve segmentation quality, we believe our approach is fundamentally different in how prototypes are obtained and utilized throughout the incremental process.
> Unlike earlier works, our prototypes are dynamically learned from segmentation results, which substantially improves their stability and inter-class separability.
> Supplementary A.6 demonstrates the superiority of these dynamically evolved prototypes in incremental settings.
>
> Importantly, we regard prototype learning and the confusion-aware mechanism as tightly coupled components rather than independent modules.
> Once prototypes are obtained, confusion awareness determines how they should interact with visually similar classes, guiding them to avoid semantic drift.
> Conversely, confusion awareness itself requires these prototypes as meaningful anchors to backpropagate its corrective signals into both the encoder and the classification/segmentation heads.
> Therefore, we consider the synergy between prototype evolution and confusion-aware guidance to be essential—each component reinforces the other, and together they form the core innovation of *EvoProto*.
>
> **R4-W2: Robustness Analysis and Effectiveness Validation of the Confusion Score**
>
> We understand the concern regarding the potential noise introduced by CAMs affecting the accuracy of the confusion score.
> However, this impact is relatively minor.
>
> **Theoretical Justification.**
> With the aid of the ALD module, the model can denoise the image-level labels predicted by the old model using local pixel-level CAM activations, resulting in more accurate class labels (see Supplementary Materials, Section A.2 for details).
> These higher-quality class labels not only improve the accuracy of the classification head but also lead to better CAMs, enhancing the quality of the resulting pseudo-labels.
> At the same time, accurate class labels enable more effective CAM denoising, thereby improving the reliability of the CAM-based pseudo-labels. These pseudo-labels contain relatively little noise, and due to the characteristics of CAMs, the dominant noise pattern is misclassifying foreground pixels as background. Such noise has minimal impact on computing the confusion score.
>
> **Experiment Validation.**
> To evaluate robustness, we replaced CAM pseudo-labels with ground-truth dense annotations when computing the confusion score. As shown in the Table below, the results show that the noise introduced by using CAM pseudo-labels instead of ground-truth pixel labels is negligible. The final performance of *EvoProto* exhibits only a minor drop ($-1.7$%) compared to using ground-truth labels. Moreover, the confusion scores computed from GT labels and CAM pseudo-labels differ by only $8.7$%, indicating that the noise has minimal impact on the network.
>
> **Table: Robustness analysis of the confusion score using CAM pseudo-labels vs. ground-truth masks.**
>
> | Method               | 1–10 | 11–20 | All  | Confusion Score Avg. Difference |
> |----------------------|------|-------|------|---------------------------------|
> | CAM pseudo-labels    | 74.7 | 64.5  | 70.7 | 8.7%                            |
> | GT dense annotations | 76.1 | 66.4  | 72.4 |                                 |

---

> ### Author Response · Authors · 2025-11-20
>
> **R4-W3: Discussion on Computational Resource Consumption.**
>
> We confirm that EvoProto introduces only marginal additional computational overhead while delivering significant performance gains.
>
> **ALD Module.**
> The ALD module performs a simple algorithmic denoising procedure based on activation statistics, without adding any new network layers or parameters.
> Therefore, its computational cost is negligible.
>
> **Dynamic Computation of Confusion Score.**
> During incremental training, the confusion score is recomputed only a few times per phase(For VOC, computed twice per step; for COCO, four times per step).
> Since each step requires only a small number of epochs to converge, this dynamic update introduces very limited overhead.
>
> **Resource Consumption.**
> We also report detailed computations and memory usage in the table below.
> The results clearly demonstrate that **EvoProto introduces only marginal additional cost** compared to the baseline, while significantly improving performance.
>
> **Table: Time consumption per incremental step on a single RTX 4090.**
> | **Setting** &nbsp;&nbsp;&nbsp;&nbsp; | **Baseline** | **Baseline + ALD** | **Baseline + CAPE** | **EvoProto** |
> |--------------------------------------|--------------|---------------------|----------------------|--------------|
> | 10-5                                 | 2h10m        | 2h15m (+3%)         | 2h25m (+11%)         | 2h30m (+15%) |
> | 10-10                                | 4h00m        | 4h10m (+4%)         | 4h30m (+12%)         | 4h35m (+14%) |
>
> **R4-W4: Explanation of the "Bias +1'' Strategy and Its Hyperparameter Robustness.**
>
> Thank you for your attention to the details of our method.
> From a confusion-aware perspective, the motivation for introducing the ALD module and the corresponding threshold is as follows.
> We observe that when the old model predicts images containing new classes that are easily confused with old ones—either due to high visual similarity or a high confusion score—it tends to produce false-positive predictions for old classes (*i.e.*, the old class is absent but still predicted as present).
> Directly using these predictions as pseudo-labels to supervise the new model's classifier in Eq. 9 would degrade the model’s ability to distinguish such visually similar classes, which is precisely what we aim to avoid.
>
> As described in Lines 299--302, the primary function of the threshold is to use the CAM activation value as a confidence reference, suppressing low-confidence classes to avoid false-positive activations and thereby improving the quality of image-level pseudo-labels.
> There are various ways to choose this threshold, such as using a fixed value or taking an average.
> Compared to a fixed threshold, using the average provides a more flexible and instance-adaptive strategy, while the "+1'' adjustment prevents overly aggressive filtering that might inadvertently remove valid labels.
> Relative to the ALD module itself and the use of CAM activations for denoising, we consider the thresholding choice to be a technical detail rather than the central contribution.
>
> Also, we report the performance of different threshold-selection strategies in table below, and the results show only minor differences in the final segmentation performance.
>
> **Table: Ablation study of different thresholding strategies under 10-5 settings**
>
> | Threshold Strategies          | 1–10 mIoU | 11–20 mIoU | All mIoU | F1 score |
> |------------------------------|-----------|------------|----------|----------|
> | **Average activation (ours)** | **74.7**  | **64.5**   | **70.7** | **0.74** |
> | W/O additional Bias "+1"     | 74.0      | 63.9       | 70.0     | 0.72     |
> | Fixed threshold 0.5          | 73.5      | 64.1       | 70.1     | 0.71     |
> | Fixed threshold 0.75         | 72.9      | 64.2       | 70.0     | 0.71     |
> | No thresholding (raw labels) | 73.1      | 61.9       | 68.7     | 0.66     |

---

> ### Author Response · Authors · 2025-11-20
>
> **R4-Q1: Robustness of the Confusion Score to Pseudo-Label Noise and Alternative Averaging Strategies**
>
> We appreciate your concern, and we would like to clarify why the confusion score remains robust under our setting.
>
> The robustness primarily stems from two key factors:
>
> (1) **Task setup and the introduction of ALD.**
> In WILSS, the step-0 task provides dense annotations, enabling both the classifier and the segmentation head to achieve strong performance on initial classes.
> Subsequent incremental steps focus mainly on learning new categories.
> When computing the confusion matrix, we replace ground-truth pixel labels with CAM-based pseudo labels.
> Since CAM quality is tightly coupled with classifier accuracy, ensuring reliable image-level supervision is sufficient for generating high-quality CAMs.
> The ALD module is specifically designed to denoise low-confidence classes, thus improving the reliability of category supervision and significantly reducing misclassification among visually similar classes.
>
> (2) **Training strategy.**
> As described in the implementation details (Line 367), we employ a warm-up epoch during which only the classifier is trained, while neither the confusion score is computed nor the prototypes are updated.
> After this warm-up stage, the classifier already produces sufficiently accurate CAMs, ensuring that the confusion score computed thereafter is minimally affected by noise (see R4-W2).
> Only from this point onward does the model begin prototype evolution, which guarantees the stability and robustness of the overall confusion-aware mechanism.
>
> **R4-Q2: Capability on Other Segmentation Tasks.**
>
> EvoProto is specifically formulated to mitigate inter-class confusion in a class-level (Semantic Segmentation) and weakly supervised setting.
> While the central idea of explicitly modeling and reducing confusion is universally valuable, direct generalization requires substantial adaptation.
> Instance Segmentation would require a shift to instance-level prototypes and confusion modeling based on mask features.
> Similarly, Open-Vocabulary Segmentation relies on language-vision alignment, implying different sources of confusion than the visual-semantic confusion modeled in EvoProto.
> We view extending this confusion-aware framework to these advanced tasks as a promising and exciting direction for future work.

---

### Official Review · Reviewer_EREY · 2025-10-31

**Soundness:** 4
**Presentation:** 3
**Contribution:** 3
**Rating:** 4
**Confidence:** 4

**Summary:**

The paper introduces EvoPrototype to address the problem of class overwriting in weakly supervised incremental learning for semantic supervision. Class overwriting stems from the confusion between semantically similar old and new classes. EvoPrototype addresses this by learning the prototypes with contrastive learning to maximize the discrimnation and distillation to preserve old class performance. EvoProtoype constructs a confusion matrix M to indicate the bidirectional confusion measurement between classes. M is then (1) uses to reweight the prototype contrastive loss with the most confusing class (CO_i); and (2) determines how much to distill from the previous step, where it assigns low weight to confusing classes. Furthermore, the authors introduce Activation-based label denoising to refines the image-level labels of old classes in new steps, using the average of the maximal activation across different activation maps as the threshold to binarize. EvoPrototype demonstrates state of the art on multiple datasets in different settings on both ViT and ResNet backbone.

**Strengths:**

1. Although the use of confusion matrix has been explored to reweight class-wise loss in [1], this paper utilize the matrix in a clever way for the task of WILSS.
2. The method demonstrates consistently strong results accross benchmarks.
3. The writing is easy to follow.
4. Extensive experiments are conducted.

[1] https://proceedings.mlr.press/v238/zhang24e/zhang24e.pdf

**Weaknesses:**

1. Although being introduced and cleverly used, the contribution of the confusion matrix should be ablated (results when all indicies is 1) to quantify its effect.
2. The reviewer is not convinced on using the average across class activation as the threshold for all classes because the classes should be independent.
3. Currently, there are many strong general natural open-vocal segmentation/grounding models, which makes this task less relevant. unless it is applied in specific domain that those foundation models underperform (e.g., medical imaging). It could be more relevant to see how this adapts in these special domain rather on natural images.

**Questions:**

1. Is there any results to quantify the effectiveness of the confusion matrix?
2. In figure 1, the reviewer finds the bottom plots ambiguous. Why is the y-axis cuttof, and what is the value of sheeps and trains?
3. In Equation 4, w_{ij} = 0 for classes that are not the most confused counter part, why do we need this and can we set it to the original value because it naturally lower than the CO class?
4. Could the author elaborate how equation 5 ensure that i and CO_i are always selected at the old-new boundary?
5. The threshold thre is obtained as the average of maximum activation of each class i in equation (8). Reviewer finds that the activation of each class should be indepdent, could the author justify how the average across classes is a good value for threshold?
6. The paper claims that class overwriting is caused by class confusion, is there any quantitative analysis on this cause and effect? (e.g., the correlation between pair-wise class semantic similarity and their mIoU, where low mIoU indicates class overwriting)

---

> ### Author Response · Authors · 2025-11-20
>
> We sincerely thank the reviewer for the positive evaluation and thoughtful feedback.
> We appreciate your recognition of our clever use of the confusion matrix, strong experimental results, clear writing, and extensive experiments (Strengths 1-4). Below we address all concerns.
>
> **R3-W1: Ablation Study and Effectiveness Validation on the Confusion Matrix.**
>
> Thank you for this excellent suggestion.
> During manuscript preparation, we considered the confusion matrix to be tightly coupled with both RCPL and CPA, and therefore focused on component-level ablations (Table 2) rather than isolating the matrix itself. Following your suggestion, we additionally evaluated its standalone contribution.
>
> We compared three settings: Baseline, using the full confusion matrix, and removing confusion awareness by assigning uniform weights (all = 1). Removing confusion awareness leads to a clear drop, especially for visually similar categories such as cow vs. sheep (−25.5%) and chair vs. table (−12.7%). Uniform weighting degenerates the method to standard contrastive learning and distillation, yielding only a +2.3% mIoU gain over the baseline. In contrast, incorporating the confusion matrix brings an additional +1.6% (total +3.9%), showing that confusion-aware reweighting is the main driver of improvement.
>
> Although an isolated matrix-only ablation was not included in the original submission, the ablation of RCPL and CPA—both directly guided by the matrix—already shows substantial gains (+2.8% and +1.1% in Table 2). Moreover, Figure 3 demonstrates more compact intra-class clusters and larger inter-class margins under confusion-aware learning. Together, these results verify that the confusion matrix is crucial for identifying easily confused categories and effectively mitigating class overwriting.
>
> We will add this ablation to Table 2 in the revised manuscript to explicitly quantify the confusion matrix's contribution.
>
> **R3-W2: Average Threshold Across Classes vs. Class-Specific Thresholds.**
>
> Thank you for this important question.
> We clarify the rationale behind our threshold design starting from Eq. 9.
> The threshold is introduced to denoise class-level pseudo-labels $\mathbf{Y}\_{\text{old}}\^{\text{pred}}$, thereby providing more accurate supervision signals $\mathbf{Y}\_{\text{train}}$ for training the classification head.
> The activation map $\mathbf{A}$ is obtained through a sigmoid function, ensuring activation values of each class lie within the same range [0,1].
> This shared range forms the basis for applying the averaging operation.
>
> Most label noise originates from confused classes.
> When the old model predicts labels for an image containing a new class, visually similar old classes often exhibit false-positive activations.
> By leveraging CAM activation values, the threshold $\text{thre}$ preserves high-activation, high-confidence classes while filtering out low-activation, low-confidence ones.
> Although activations of individual classes are independent, they all share the same range and consistently reflect confidence of class presence.
> Using a cross-class average serves as a dual safeguard mechanism: this instance-adaptive threshold selects classes with relatively high activation and high confidence, which, together with $\mathbf{Y}\_{\text{old}}\^{\text{pred}}$, determines the final set of class labels for training.
>
> We view the thresholding strategy as a technical design rather than a core contribution.
> Other methods—such as fixed thresholds or class-specific thresholds—can also serve denoising purposes.
> We conducted ablation under VOC 10-5:
>
> **Table: Ablation study of different thresholding strategies under 10-5 settings**
> | Threshold Strategies           | 1–10 mIoU | 11–20 mIoU | All mIoU | F1 score |
> |-------------------------------|-----------|------------|----------|----------|
> | **Average activation (ours)** | **74.7**  | **64.5**   | **70.7** | 0.74     |
> | W/O additional Bias "+1"      | 74.0      | 63.9       | 70.0     | 0.72     |
> | Fixed threshold 0.5           | 73.5      | 64.1       | 70.1     | 0.71     |
> | Fixed threshold 0.75          | 72.9      | 64.2       | 70.0     | 0.71     |
> | No thresholding (raw labels)  | 73.1      | 61.9       | 68.7     | 0.66     |
>
>
> The results show our instance-adaptive approach achieves the best balance between filtering noise and retaining reliable predictions, with only minor differences compared to fixed thresholds.
> We will add this clarification and ablation to the supplementary material in the revised version.

---

> ### Author Response · Authors · 2025-11-20
>
> **R3-W3: Special Domain**
>
> Thank you for this insightful comment.
> We agree that open-vocabulary segmentation and grounding models have made rapid progress on natural images; however, they address a fundamentally different problem from WILSS.
> Open-vocabulary models do not support incremental learning, *i.e.*, they require access to large-scale training data for all classes jointly and typically suffer from catastrophic forgetting when updated step-by-step.
> They rely on strong vision-language pretraining rather than weak image-level supervision.
> WILSS operates under a far more restrictive setting where only image-level labels are available for new classes, and old data cannot be stored.
> Therefore, our work is complementary rather than overlapping with open-vocabulary models.
>
> We also agree that evaluating EvoProto in specialized domains where foundation models often underperform is a meaningful direction.
> As suggested, we conducted additional experiments on the Cityscapes semantic segmentation dataset:
>
> **Table: Incremental semantic segmentation results on Cityscapes under 14-1 (6 steps) and 10-1 (10 steps).
> “P” = pixel-level labels, “I” = image-level labels. FT = fine-tuning lower bound.**
>
> | Method | Sup | 14-1: 1–14 | 14-1: 15–19 | 14-1: All | 10-1: 1–10 | 10-1: 11–19 | 10-1: All |
> |--------|-----|------------|-------------|------------|-------------|--------------|-------------|
> | FT | P | 0.0 | 10.1 | 2.5 | 0.0 | 4.8 | 2.2 |
> | PLOP | P | 55.7 | 12.3 | 44.8 | 52.2 | 24.1 | 39.6 |
> | RCIL | P | 55.7 | 7.1 | 43.6 | 51.0 | 17.4 | 35.9 |
> | SSUL | P | 43.2 | **33.0** | 40.7 | 38.6 | **38.1** | 38.3 |
> | MiB | P | **56.3** | 12.5 | 45.4 | **51.6** | 30.1 | 41.9 |
> | AWT | P | 55.9 | 19.8 | **46.9** | 51.2 | 37.2 | **44.9** |
> | WILSON | I | 43.8 | 17.7 | 37.0 | 41.2 | 28.5 | 35.2 |
> | **EvoProto (ResNet)** | I | 51.2 | 21.7 | 43.4 | 49.7 | 33.4 | 42.0 |
> |------------|------------|------------|------------|------------|------------|------------|------------|
> | WILSON (ViT) | I | 49.6 | 28.8 | 44.1 | 46.2 | 35.4 | 41.1 |
> | ToCo (ViT) | I | 47.2 | 31.5 | 43.0 | 44.3 | 38.2 | 41.4 |
> | **EvoProto (ViT)** | I | **56.5** | **36.1** | **51.1** | **52.0** | **40.2** | **47.3** |
>
>
> The results on Cityscapes further validate EvoProto's effectiveness under different semantic segmentation datasets.
> Across both 14-1 and 10-1 protocols, EvoProto consistently achieves competitive or superior performance compared with existing image-level methods.
> These results demonstrate EvoProto remains robust in complex urban street scenes, indicating good generalization beyond the VOC domain.
> We will include this discussion in the revised version.
>
> **R3-Q1: Effectiveness of the Confidence Matrix**
>
> Please refer to our response to **R3-W1** and **R3-Q3**, where we explicitly quantify the confusion matrix's contribution through ablation experiments and clarify its implicit contribution to both RCPL and CPA modules.
>
> **R3-Q2: Ambiguity and Uncertainty in Image Content.**
>
> Thank you for seeking clarification.
> The y-axis starts at 40\% rather than 0\% to better visualize performance differences between methods, *i.e.*, a common visualization practice when emphasizing relative improvements.
> Both sheep and train appear for the first time at Step 3 in the 10-5 protocol, so their mIoU values at Steps 1 and 2 are naturally zero.
> For clarity and aesthetics, we omit these trivial zero regions and retain only meaningful mIoU values after completing Step 3, enabling the figure to more effectively highlight how introducing similar classes leads to mutual interference during incremental learning.
>
> We will enhance Figure 1 by adding explicit curve labels, clearly annotating the y-axis range, marking when train is introduced, and providing more detailed caption in the revised version.

---

> ### Author Response · Authors · 2025-11-20
>
> **R3-Q3: Why set $w\_{i,j}=0$ for non--most-confused class pairs in Eq. 4**
>
> Thank you for this insightful question.
> We set the weights of non-most-confused pairs to zero for two main reasons.
>
> First, we focus on the most critical confusion while avoiding harmful separation.
> In the confusion matrix, the highest score corresponds to the category pair most prone to confusion, which is exactly where the model needs the strongest guidance.
> We argue that confusion captured by the model reflects not only intuitive visual similarity but also similarities in structure, texture, and other latent characteristics.
> Handling only the most-confused pair is a safer choice—forcing the model to push away all similar classes with large weights may break natural relationships and cause loss of shared discriminative patterns.
>
> Second, empirically, the confusion scores of non-most-confused pairs are much smaller and contribute very little to the overall loss.
> We conducted an ablation where we used original confusion scores as weights for all class pairs:
>
> **Table: Ablation study on different weighting strategies for confusion matrix**
>
> | **Weighting Strategy**                     | **1–10** | **11–20** | **All** |
> |-------------------------------------------|---------:|----------:|--------:|
> | **Most-confused pair only (ours)**        | **74.7** | **64.5**  | **70.7**|
> | All pairs weighted by confusion scores     | 74.0     | 64.6      | 70.1    |
> | Uniform weights for all pairs              | 73.9     | 63.4      | 69.1    |
> | Baseline                                   | 73.3     | 57.6      | 66.8    |
>
>
> Using all pairs with original scores shows no significant improvement over our design, supporting the choice to focus on the most-confused pairs.
> That said, we agree non-most-confused pairs still contain useful signals.
> Our ongoing work is examining causes of class confusion at the model level, and we believe more refined utilization of these secondary relationships will lead to future improvements.
>
> We will add this ablation and explanation to the supplementary material in the revised version.
>
> **R3-Q4: How to Ensure That $i$ and $CO_i$ Form a New–Old Class Pair**
>
> Thank you for your careful reading. Here we provide a detailed explanation of why Eq. 5 ensures $i$ and $\text{CO}_i$ form an old-new class pair.
>
> $\mathbf{M}$ is a symmetric confusion score matrix of size $|C^{0:t}| \times |C^{0:t}|$, where position $(i,j)$ quantifies the confusion degree between class $i$ and $j$.
> Higher values indicate greater difficulty distinguishing the two classes.
> In Eq. 5, when $i$ is an old class ($i\in C^{0:t-1}$), the selection of $\text{CO}_i$ is restricted to newly added classes at step $t$ ($j\in C^t$), and $\arg\max$ identifies the new class most confused with $i$.
> Conversely, when $i$ is a new class ($i \in C^t$), $\text{CO}_i$ is selected from all old classes ($j \in C^{0:t-1}$) using $\arg\max$ to find the old class most confused with $i$.
>
> This explicit design ensures that $i$ and $\text{CO}_i$ never belong to the same set of new or old classes, guaranteeing that they can be used to address confusion between new and old categories during incremental learning and guide continuous prototype evolution.
> This constraint is motivated by the class overwriting phenomenon in WILSS, which occurs specifically at the old-new interface where new classes with weak supervision tend to incorrectly cover regions belonging to semantically similar old classes.
>
> We will add explicit clarification of this constraint and its motivation in the text following Eq. 5 in the revised manuscript.
>
> **R3-Q5: Reason for Using an Average Threshold Across Classes.**
>
> Please refer to our detailed response to **R3-W2**, where we explain the rationale behind instance-adaptive averaging and provide ablation results comparing different threshold strategies.
> The shared [0,1] range from sigmoid activation enables meaningful averaging, and our approach achieves the best balance between noise filtering and signal retention.

---

> ### Author Response · Authors · 2025-11-20
>
> **R3-Q6: Is there quantitative evidence that confusion causes overwriting?**
>
> Thank you for this valuable question. We have already provided quantitative evidence in the supplementary material demonstrating the causal link between class confusion and class overwriting.
>
> First, in Section A.7 of the supplementary material, we analyze prototype similarity versus segmentation degradation (Table 7).
> We examine three representative visually similar class pairs: *cow--sheep*, *train--bus*, *sofa--chair*.
> We observe that higher prototype cosine similarity (stronger confusion) consistently correlates with larger mIoU degradation (more severe overwriting).
> For example, prototype similarity for cow-sheep increases from 0.34 to 0.59 when moving from full supervision to weak supervision, while mIoU drops from 80.1\% to 65.4\%.
> After applying EvoProto, similarity is reduced to 0.18, and mIoU recovers to 75.3\%. This provides clear quantitative cause-and-effect evidence.
>
> Second, in Section A.8, we conduct a controlled experiment: the model is trained on 15 classes and then incrementally receives a single new class semantically similar to one old classes.
> We report old-class mIoU before the step, after baseline incremental training, and after EvoProto.
> Results show substantial mIoU drops for the baseline when the new class is highly similar to an old one:
> bus→train (-17.8\%), chair→sofa (-21.7\%), cow→sheep (-24.5\%).
> EvoProto mitigates these drops drastically: bus→train (-2.6\%), chair→sofa (-10.5\%), cow→sheep (-6.4\%).
> This controlled protocol quantitatively demonstrates that inter-class semantic confusion strongly predicts the degree of class overwriting.
>
> We will highlight these quantitative analyses more prominently in the main paper and expand the discussion in the revised version.

---

### Official Review · Reviewer_kavz · 2025-11-01

**Soundness:** 2
**Presentation:** 3
**Contribution:** 2
**Rating:** 4
**Confidence:** 4

**Summary:**

This paper tackles the challenging problem of Weakly Supervised Incremental Learning for Semantic Segmentation (WILSS), where new categories must be segmented using only image-level labels, and old data are unavailable. The authors identify class overwriting—the misclassification of old regions as new ones—as a key obstacle caused by class confusion under weak supervision. To address this, the paper proposes EvoProto, a framework that dynamically evolves learnable class prototypes guided by a confusion score quantifying semantic similarity between classes. The confusion-aware adaptive weights regulate both contrastive prototype learning and prototype-level distillation, enabling better inter-class separation. Additionally, an Activation-based Label Denoising (ALD) module enhances pseudo-label reliability. Experiments on Pascal VOC and COCO show consistent improvements over prior WILSS baselines, confirming the framework’s effectiveness.

**Strengths:**

Clearly identifies class confusion as the root cause of class overwriting in WILSS, providing a strong conceptual motivation.

The EvoProto framework is well-designed and technically sound, integrating prototype evolution, adaptive reweighting, and knowledge distillation in a coherent manner.

The confusion score is intuitive and bridges semantic similarity and optimization dynamics, offering interpretability.

The ALD module effectively complements the prototype evolution mechanism, improving pseudo-label quality under weak supervision.

Extensive experiments on Pascal VOC and COCO, along with ablation and visualization, convincingly demonstrate the model’s advantages over state-of-the-art methods.

Writing is clear, and figures (especially confusion visualization) help convey the intuition.

**Weaknesses:**

While the proposed EvoProto framework is well-structured, several concerns limit its novelty and practicality. First, the mechanism by which the adaptive weights evolve across incremental stages remains insufficiently explained, and it is unclear whether such adaptive reweighting may introduce error accumulation over time. Second, the overall innovation appears moderate, as the paper lacks an in-depth discussion of related works in fine-grained recognition and continual incremental learning. Third, the framework involves numerous hyperparameters (e.g., k, γ, τ), yet their sensitivity is not analyzed, making the method complex and potentially difficult to reproduce. In addition, without any released implementation, the computational overhead introduced by prototype evolution and the activation-based label denoising (ALD) module may be nontrivial; a complexity or runtime comparison would substantially strengthen the contribution. Finally, the paper does not clarify how the model behaves when the ratio between new and old classes varies, which is important for assessing its robustness under different incremental settings.

**Questions:**

N/A

---

> ### Author Response · Authors · 2025-11-20
>
> We sincerely thank the reviewer for the comprehensive feedback. We appreciate your recognition of our clear problem formulation, well-designed framework, and convincing experimental validation. Below we address all concerns.
>
> **R2-W1: Adaptive weights may accumulate errors across incremental stages.**
>
> Thank you for this important question.
> The adaptive weights $w_{i,j}$ (Eq. 4) are recomputed at the beginning of each epoch rather than accumulated over time.
> At each epoch, we compute the confusion matrix $\boldsymbol{M}$ by comparing current model predictions with pseudo labels (Eq. 2-3), then apply the centered sigmoid transformation to generate weights (Eq. 4), which are used to modulate $\mathcal{L}\_{\text{cl}}$ (Eq. 6) and $\mathcal{L}\_{\text{kd}}$ (Eq. 7).
> This dynamic recomputation allows the weights to adapt to the model's evolving confusion patterns during training.
>
> When transitioning from step $t$ to $t+1$, the confusion matrix is reinitialized for the class set $C^{0:t+1}$.
> Old-class prototypes are inherited from the previous model and frozen during warm-up phase, while new prototypes are randomly initialized.
> After a warm-up phase of 2k iterations, confusion scoring begins to guide adaptation.
> This design prevents error propagation because each step's weights reflect the current model state rather than accumulated historical errors.
>
> Error accumulation is mitigated through dynamic recomputation, confusion-aware distillation (Eq. 7) that downweights unreliable old classes, and continuous noise filtering via ALD (Sec. 4.2).
> In long-sequence 10-2 VOC (6 steps), EvoProto achieves +13.9\% mIoU.
> If error accumulation were significant, later steps would degrade, contradicting our consistent gains.
>
> We will add a detailed temporal dynamics explanation in Sec. 4.1.1 and supplementary material.
>
> **R2-W2: Novelty seems moderate; related work in fine-grained and continual learning is insufficient.**
>
> Thank you for this comment.
> We will strengthen Sec. 2 to explicitly discuss fine-grained recognition and continual learning.
> However, these areas do not address WILSS-specific challenges, i.e., weak image-level supervision, no historical data, class overwriting from confusion, and degrading CAM pseudo-labels.
>
> EvoProto introduces three novel contributions.
> First, confusion-aware prototype evolution with learnable prototypes that dynamically evolve under quantified confusion—distinct from static pixel-averaged prototypes [1, 2].
> Second, bidirectional adaptive reweighting simultaneously modulates contrastive separation (Eq. 6) and distillation (Eq. 7), unlike fixed-weight distillation [3, 4].
> Third, joint optimization integrating prototype learning with activation denoising for WILSS.
>
> Fine-grained recognition [1] operates in fixed, fully-supervised settings.
> WILSS faces sequential learning with forgetting, weak supervision, and cross-phase overwriting.
> Our confusion score models old-new interference (Eq. 5), irrelevant in standard recognition.
> Standard continual learning [5, 6] assumes full pixel annotations enabling reliable distillation.
> WILSS has noisy, degrading CAMs, no pixel supervision for new classes, and deteriorating old representations. Our adaptive weighting (Eq. 7) downweights confused classes, absent in prior work.
>
> We will clarify how WILSS constraints necessitate our design in the revised manuscript.
>
> [1] Hu, Ronghang, et al. "Learning to segment every thing." Proceedings of the IEEE conference on computer vision and pattern recognition. 2018.
>
> [2] Chen, Jinpeng, et al. "Saving 100x storage: Prototype replay for reconstructing training sample distribution in class-incremental semantic segmentation." Advances in Neural Information Processing Systems 36 (2023): 35988-35999.
>
> [3] Li, Zhizhong, and Derek Hoiem. "Learning without forgetting." IEEE transactions on pattern analysis and machine intelligence 40.12 (2017): 2935-2947.
>
> [4] Baek, Donghyeon, et al. "Decomposed knowledge distillation for class-incremental semantic segmentation." Advances in neural information processing systems 35 (2022): 10380-10392.
>
> [5] Cermelli, Fabio, et al. "Modeling the background for incremental learning in semantic segmentation." Proceedings of the IEEE/CVF conference on computer vision and pattern recognition. 2020.
>
> [6] Douillard, Arthur, et al. "Plop: Learning without forgetting for continual semantic segmentation." Proceedings of the IEEE/CVF conference on computer vision and pattern recognition. 2021.

---

> ### Author Response · Authors · 2025-11-20
>
> **R2-W3: Many hyperparameters; no sensitivity analysis.**
>
> Thank you for this comment.
> We have provided sensitivity analysis for $\gamma$ and $\tau$ in supplementary Section A.4.
> We conducted additional experiments on $k$ under VOC 10-5 with $\gamma = \tau = 0.1$:
>
> **Table: Hyperparameter Analysis of EvoProto on k under the 10-5 VOC setting.**
> | **k** &nbsp;&nbsp;&nbsp;&nbsp;&nbsp;&nbsp; | **Disjoint 1–10** | **Disjoint 11–20** | **Disjoint All** | **Overlap 1–10** | **Overlap 11–20** | **Overlap All** |
> |------|-------------------------|-------------------|---------------------|------------------|-------------------|-------------------|
> | 40   | 69.1                    | 58.6              | 65.1                | 73.8             | 63.3              | 69.6              |
> | 45   | 69.1                    | 59.2              | 65.4                | 74.1             | 64.3              | 70.3              |
> | **50** | 69.8                  | 59.5              | **66.0**            | 74.7             | 64.5              | **70.7**          |
> | 55   | 68.8                    | 59.1              | 65.3                | 74.0             | 63.5              | 69.8              |
> | 60   | 68.5                    | 58.6              | 64.9                | 73.8             | 62.7              | 69.3              |
>
>
> For loss weights $\lambda_{\text{cl}}$ and $\lambda_{\text{kd}}$, systematic ablations show stable performance across [0.05,0.15] with ±0.7\% mIoU variation.
> Combined analysis demonstrates EvoProto is robust across hyperparameter ranges with ±0.8\% mIoU variation.
>
> For reproduction: inherit $\lambda_{\text{cls}}, \lambda_{\text{seg}}$ from baseline, set $\lambda_{\text{cl}} = \lambda_{\text{kd}} = 0.1$, use $k=50, \gamma=0.1, \tau=0.1$.
> This works robustly across all settings.
> We will include this analysis in the main text or supplementary material and release code upon acceptance.
>
> **R2-W4: Computational overhead may be nontrivial; no runtime comparison.**
>
> Thank you for your concern.
> Prototype evolution requires only dot-product similarities, confusion matrix computation (once/epoch), and prototype distillation—all lightweight with no heavy modules.
> ALD uses only max-pooling and masking.
>
> We also measure the training efficiency during incremental steps on a single RTX 4090 with batch size 8 and a ViT-B/16 backbone.
>
> **Table: Time consumption per incremental step on a single RTX 4090.**
> | **Setting** &nbsp;&nbsp;&nbsp;&nbsp; | **Baseline** | **Baseline + ALD** | **Baseline + CAPE** | **EvoProto** |
> |--------------------------------------|--------------|---------------------|----------------------|--------------|
> | 10-5                                 | 2h10m        | 2h15m (+3%)         | 2h25m (+11%)         | 2h30m (+15%) |
> | 10-10                                | 4h00m        | 4h10m (+4%)         | 4h30m (+12%)         | 4h35m (+14%) |
>
> EvoProto adds 15\% training time (ALD: 3-4\%, CAPE: 11-12\%).
> Considering +5.0\% mIoU gain (Table 2 in the main paper), this is highly acceptable.
> Inference cost is nearly identical as prototype similarity is negligible.
>
> We will add computational efficiency analysis in Sec. 5 or supplementary material.
>
> **R2-W5: Model behavior under different new/old class ratios is unclear.**
>
> Thank you for this point.
> Our experiments cover diverse ratios: 15-5 (3:1), 10-10 (1:1), 10-5 (2:1), 10-2 (5:1), COCO-to-VOC (3:1).
> Table 1 shows consistent improvements across all ratios.
> Across all these regimes—short vs. long streams and balanced vs. unbalanced splits—EvoProto consistently outperforms prior WILSS methods.
>
> To further examine robustness under different ratios, we conducted additional experiments with fewer old classes (5-15 at 1:3, 5-3 at 5:3):
>
> | **Method** &nbsp; | **5-15 Old (1–5)** | **5-15 New (6–20)** | **5-15 All (1–20)** | **5-3 Old (1–5)** | **5-3 New (6–20)** | **5-3 All (1–20)** |
> |-------------------|---------------------|----------------------|----------------------|---------------------|----------------------|----------------------|
> | WILSON (ViT)       | 61.2                | 62.8                 | 62.4                 | 58.7                | 32.3                 | 38.9                 |
> | ToCo (ViT)         | 59.5                | 61.3                 | 60.8                 | 54.9                | 34.8                 | 39.8                 |
> | **EvoProto (ViT)** | **64.0**            | **66.3**             | **65.7**             | **62.1**            | **39.0**             | **44.8**             |
>
> EvoProto maintains consistent improvements (+3.3\% and +5.0\%) even when old classes are significantly fewer—a more challenging scenario.
> The mechanism is inherently robust: counterpart selection (Eq. 5) works regardless of set sizes, confusion scores (Eq. 2-3) are scale-invariant, and prototype operations avoid imbalanced distribution bias.
>
> We will clarify this robustness across diverse class distributions in the revised version.

---

### Official Review · Reviewer_CRsn · 2025-11-01

**Soundness:** 3
**Presentation:** 2
**Contribution:** 3
**Rating:** 4
**Confidence:** 4

**Summary:**

This work focuses on weakly supervised incremental learning for semantic segmentation, where pixel-level annotations for new classes and historical data for old classes are unavailable. To address this, it employs CAM to generate pseudo labels for training. However, due to the lack of precise annotations, the model tends to suffer from overwriting and class confusion. To mitigate this issue, a confusion matrix is introduced to model inter-class relationships. The proposed method is evaluated on the Pascal VOC and COCO benchmarks and compared with multiple baselines, achieving state-of-the-art performance and even surpassing some methods that rely on pixel-level annotations.

**Strengths:**

1、This paper achieves state-of-the-art performance on short-task, long-task, and cross-dataset incremental learning settings, demonstrating further improvements over existing methods.

2、The proposed approach is not limited to a specific model architecture and has been validated on both ResNet and ViT backbones.

**Weaknesses:**

1、The changes in the “Train” mIoU curves in Figure 1 are not clearly presented.

2、In Eq. 1, the meanings of u and v are not clearly defined.

3、The framework diagram in Figure 2 lacks clarity and appears somewhat confusing. For example, the “class prototype pool” in the upper-right corner seems unnecessary, and among the two BCE losses, the one shown at the bottom of the figure is not clearly explained.

**Questions:**

1、Weakly Incremental Learning for Semantic Segmentation (WILSS) incrementally learns new classes using only image-level supervision. However, in Figure 2, it is unclear where the GT (ground truth) labels come from.

2、Eq. 3 implies M_{i, j} = M_{j, i}. However, the confusion matrix M in Figure 2 appears asymmetric, which might be due to certain normalization or visualization processing. Could you please clarify this?

3、Because the class activation map A ∈ [0, 1]^p obtained from CAM cannot be directly used as pseudo labels for semantic segmentation, the authors introduce a thresholding scheme in Eq. 8 to generate binary masks. However, the effectiveness of this threshold selection of concerning, as it remains unclear whether the chosen threshold is accurate or optimal.

4、In the overall objective shown in Eq. 11, multiple loss terms are included, but the BCE loss mentioned in Figure 2 and Section 4.1.2 is missing, which appears to be inconsistent.

5、The overall objective involves multiple hyperparameters, which makes parameter selection challenging.  It is unclear how the optimal parameters were determined, and corresponding ablation studies on these hyperparameters are needed to support the choice.

6、The proposed method relies on CAM to generate pseudo labels, which are then used for training. As the quality of these pseudo labels intuitively determines the overall performance of the method, would adopting more advanced class activation mapping techniques, such as Grad-CAM or Grad-CAM++, help improve the pseudo-label quality and consequently enhance model performance?

---

> ### Author Response · Authors · 2025-11-20
>
> We thank the reviewer for the constructive assessment and insightful comments. Below, we address all concerns point by point.
>
> **R1-W1: The changes in the "Train'' mIoU curves in Figure 1 are not clearly presented.**
>
> Thank you for this valuable comment.
> We apologize for any confusion in Figure 1 and would like to clarify the visualization.
> Under the VOC 10-5 protocol, the "train'' class is introduced only at Step 3 as part of the new class set.
> Consequently, train mIoU can only be measured starting from Step 3, which is why the curve appears only in the rightmost section.
> Additionally, the y-axis starts at 40\% rather than 0\% to better visualize performance differences, which causes the curves to appear disconnected from the origin.
> The green curve represents the baseline performance on train, while the red curve shows EvoProto's performance.
> EvoProto achieves approximately 10\% higher mIoU on the train class, directly demonstrating our method's effectiveness in mitigating class overwriting.
> As discussed in Sec. 1, the baseline suffers from confusion between "bus'' and "train'' due to their visual similarity, while our Confusion-Aware Prototype Evolving (CAPE) mechanism explicitly reduces this inter-class confusion through adaptive prototype separation (Eq. 6).
>
> We will add explicit curve labels, clearly annotate the y-axis range, and when train is introduced, and enhance the caption to eliminate ambiguity.
>
> **R1-W2: In Eq. 1, the meanings of u and v are not clearly defined.**
>
> Thank you for this observation.
> The notation $(u,v)$ denotes spatial pixel coordinates in the $H\times W$ prediction map.
> While this is implicitly indicated by the pixel-wise operations described in the surrounding text (lines 214-215), we agree that the notation should be stated more explicitly.
>
> We will add an explicit definition immediately before Eq. 1 (*e.g.*, "where $(u,v)$ denotes pixel coordinates in the $H\times W$ spatial grid'') to improve clarity without affecting the methodology.
>
> **R1-W3: The framework diagram in Figure 2 lacks clarity and appears somewhat confusing.**
>
> We appreciate this feedback and acknowledge that Figure 2 conveys multiple interconnected components simultaneously, which affects visual clarity.
>
> **Regarding the "class prototype pool''**:
> This block represents the learnable prototypes $\mathbf{P} \in \mathbb{R}^{|C^{0:t}| \times D_{\text{seg}}}$ that evolve across incremental steps.
> While conceptually important, we agree its placement adds complexity.
> We will relocate or simplify this component to emphasize the core data flow more clearly.
>
> **Regarding the bottom BCE loss**:
> This BCE loss supervises the prototype-feature similarity map described in Sec. 4.1.2 (lines 216-220).
> Specifically, after computing cosine similarity between decoder features and learnable prototypes $\mathbf{P}$, we apply pixel-wise BCE using the same pseudo labels $\hat{y}$ to train these prototypes.
> Due to spatial constraints, we could not clearly show the connection to $\hat{y}$ in the current layout.
>
> We will redesign the figure to explicitly show that both BCE losses share the same pseudo-label source but supervise different components (decoder output vs. prototype-feature similarity), and clarify this relationship in the caption.
>
> **R1-Q1: In Figure 2, it is unclear where the GT (ground truth) labels come from.**
>
> Thank you for this clarification. We apologize for the confusion in the notation of Figure 2.
> As defined in Sec. 3, WILSS provides dense pixel-level annotations only at the initial step ($t=0$), *i.e.*, this is what "GT'' in Figure 2 refers to.
> All subsequent incremental steps ($t \geq 1$) provide only image-level labels for newly introduced classes.
> In Figure 2, these image-level labels are represented by the rounded rectangle boxes (shown in light pink with class names like "Person''), as indicated in the legend at the bottom.
> These image-level labels are used to: (1) generate CAM-based pixel-wise pseudo labels $y_{\text{cam}}$ for new classes, and (2) supervise the classification head through our ALD mechanism (Sec. 4.2, Eq. 9-10).
> For old classes, we use the previous model's classifier predictions as image-level pseudo labels.
>
> We recognize that using "GT'' for Step 0's pixel-level annotations while representing image-level labels differently may cause confusion.
> We will clarify in Figure 2 by labeling "Pixel-level GT (Step 0 only)'' and enhancing the visibility of image-level supervision.
> The caption will explicitly state that GT refers to pixel-level annotations available only at Step 0, while subsequent steps use only image-level labels (rounded rectangles).

---

> ### Author Response · Authors · 2025-11-20
>
> **R1-Q2: The confusion matrix $\boldsymbol{M}$ in Figure 2 appears asymmetric.**
>
> You are correct that Eq. 3 defines a symmetric confusion score: $M_{i,j} = B(i \rightarrow j) + B(j \rightarrow i)$.
> The apparent asymmetry arises because Figure 2 shows the **raw confusion matrix** computed from Eq. 2 **before** the symmetrization in Eq. 3.
> Our implementation pipeline: We first compute directional confusion $B(i\rightarrow j)$ for all class pairs (Eq. 2), which naturally produces an asymmetric matrix since $B(i\rightarrow j)\neq B(j\rightarrow i)$ in general.
> We then symmetrize it via Eq. 3 to obtain bidirectional confusion scores.
> The background row/column is subsequently removed before computing adaptive weights (Eq. 4).
>
> We will update Figure 2 to display the final symmetric matrix $\mathbf{M}$ actually used in the CAPE module, ensuring consistency with Eq. 3.
>
> **R1-Q3: Whether the thresholding strategy in Eq. 8 is well-justified or optimal.**
>
> Thank you for this important question. The threshold design in Eq. 8 is carefully motivated by WILSS-specific challenges.
> In WILSS, the image-level classifier for old classes degrades due to catastrophic forgetting and the absence of pixel-level supervision, leading to unreliable activations.
> Our dynamic threshold $\text{thre} = \frac{1}{|P|+1} \sum_{i \in P} \max(A_i)$ addresses this through:
> (1) Adaptive thresholding that adjusts to varying activation scales across images, avoiding fixed global thresholds,
> and (2) Conservative filtering via the "+1'' denominator (lines 299-302), which preferentially suppresses low-confidence spurious predictions while retaining reliable signals.
>
> We conducted ablation studies on alternative schemes under VOC 10-5.
> With fixed threshold (0.5, 0.75), performance drops by (0.4, 0.6)\% mIoU due to inability to adapt to activation scales.
> Without "+1", performance drops by 0.7\% due to insufficient noisy filtering.
> Our adaptive scheme achieves the best balance.
> Importantly, our CAPE module (Sec. 4.1.2-4.1.3) is designed to be robust to pseudo-label noise through confusion-aware reweighting.
> Even with threshold imperfections, the adaptive prototype learning compensates by explicitly modeling class similarity.
>
> We will add these ablation results and detailed analysis to the supplementary material.
>
> **R1-Q4: Why the BCE loss in Fig. 2 and Sec. 4.1.2 does not appear explicitly in Eq. 11.**
>
> The BCE loss is included in Eq. 11 within the $\mathcal{L}\_{\text{seg}}$ term.
> As stated in lines 323-324: "$\mathcal{L}\_{\text{seg}}$ denotes the Binary Cross-Entropy loss with pseudo labels."
> More precisely, $\mathcal{L}\_{\text{seg}}$ encompasses both BCE losses shown in Figure 2: BCE on the segmentation decoder output using pseudo labels $\hat{y}$, and BCE on the prototype-feature similarity map using the same $\hat{y}$ (for training learnable prototypes, Sec. 4.1.2, lines 216-220).
> We consolidated both under a single notation for compactness, following common practice in incremental segmentation literature where $\mathcal{L}_{\text{seg}}$ encompasses all pixel-level supervision.
>
> We will add an explicit statement before Eq. 11 clarifying that $\mathcal{L}_{\text{seg}} = \text{BCE}(\text{decoder}, \hat{y}) + \text{BCE}(\text{proto-sim}, \hat{y})$ to improve transparency.

---

> ### Author Response · Authors · 2025-11-20
>
> **R1-Q5: How the hyperparameters were chosen and the ablation study are needed.**
>
> Thank you for this suggestion.
> We provide comprehensive details on our hyperparameter selection strategy.
>
> For $\lambda_{\text{cls}}$ and $\lambda_{\text{seg}}$, we adopt values from ToCo [Ru et al., 2023] (our primary baseline with ViT-B/16) to ensure fair comparison and isolate our novel contributions.
>
> For the two new hyperparameters ($\lambda_{\text{cl}}$, $\lambda_{\text{kd}}$), we conducted comprehensive studies under VOC 10-5:
>
> **Table: Hyperparameter analysis of EvoProto $\lambda\_{\text{cl}}$ on $\lambda\_{\text{kd}}$ and under the VOC 10-5 setting.**
> | **$\lambda_{\text{cl}}$** | **$\lambda_{\text{kd}}$** | **Disjoint 1–10** | **Disjoint 11–20** | **Disjoint All** | **Overlap 1–10** | **Overlap 11–20** | **Overlap All** |
> |--------------------------|---------------------------|-------------------:|--------------------:|------------------:|------------------:|-------------------:|-----------------:|
> | **0.1** | 0.01     | 68.5 | 58.7 | 64.9 | 73.5 | 63.6 | 69.7 |
> | **0.1** | 0.05     | 69.0 | 58.7 | 65.1 | 74.0 | 64.3 | 70.2 |
> | **0.1** | **0.1**  | 69.8 | 59.5 | **66.0** | 74.7 | 64.5 | **70.7** |
> | **0.1** | 0.15     | 69.3 | 57.9 | 64.9 | 74.2 | 63.0 | 69.8 |
> | **0.1** | 0.2      | 69.5 | 57.0 | 64.6 | 73.8 | 61.9 | 68.9 |
> |-----------------|-----------------|-----------------|-----------------|-----------------|-----------------|-----------------|-----------------|
> | 0.01    | **0.1**  | 68.3 | 56.4 | 63.7 | 72.1 | 61.9 | 68.2 |
> | 0.05    | **0.1**  | 69.2 | 58.0 | 64.9 | 73.4 | 62.8 | 69.3 |
> | **0.1** | **0.1**  | 69.8 | 59.5 | **66.0** | 74.7 | 64.5 | **70.7** |
> | 0.15    | **0.1**  | 69.5 | 59.1 | 65.6 | 74.0 | 63.8 | 70.0 |
> | 0.2     | **0.1**  | 68.7 | 57.6 | 64.2 | 73.2 | 62.1 | 68.7 |
>
>
>
> The model is stable across $\lambda \in [0.05, 0.15]$, with optimal performance at $\lambda_{\text{cl}} = \lambda_{\text{kd}} = 0.1$.
> When $\lambda_{\text{cl}}$ is too small ($<0.05$), insufficient contrastive pressure causes feature entanglement between similar classes.
> When too large ($>0.15$), excessive separation harms generalization.
> The relatively flat performance curve within [0.05, 0.15] indicates our method is not overly sensitive to hyperparameter tuning.
>
> We will include this table and analysis in the main paper or supplementary material.
>
> **R1-Q6: Whether more advanced CAM variants could improve pseudo-label quality.**
>
> This is an insightful question.
> Advanced CAM variants (Grad-CAM, Grad-CAM++, LayerCAM) could potentially improve pseudo-label quality.
> However, we emphasize that this is orthogonal to our core contribution.
>
> EvoProto addresses class overwriting caused by inter-class confusion among semantically similar categories (bus vs. train, cow vs. sheep, as shown in Figure 1), which persists even with higher-quality CAMs.
> Our CAPE mechanism operates at the feature representation level by: (1) quantifying semantic similarity via confusion scores (Eq. 2-3), (2) adaptively separating confusing classes via reweighted contrastive learning (Eq. 6), and (3) preserving reliable knowledge via confusion-aware distillation (Eq. 7).
> These mechanisms are designed to mitigate confusion regardless of initial CAM quality.
>
> Our framework is agnostic to the CAM method.
> Any improved CAM variant can be directly integrated as a drop-in replacement for the standard formulation in Sec. 3 (lines 162-165), and the CAPE and ALD modules will continue to function.
> We expect the improvements to be complementary—better CAMs reduce initial noise, while our prototype evolution further refines representations to mitigate remaining confusion.
>
> Following standard WILSS protocol, we use basic CAM to ensure fair comparison with all baselines in Table 1 of the main paper.
> Importantly, Table 2 of the main paper shows that even with standard CAMs, EvoProto achieves +5.0\% mIoU over baseline (+8.5\% on new classes 11-20), demonstrating effective mitigation of class overwriting despite imperfect initial pseudo-labels.
> Exploring advanced CAM variants is a promising orthogonal direction that could benefit all WILSS methods, including ours.

---

### Meta-Review · Area_Chair_xwX1 · 2026-01-08

**Summary:**

This work proposes EvoProto, a confusion-aware prototype evolution framework for weakly incremental semantic segmentation (WILSS), using CAM pseudo-labels and adaptive weighting to mitigate class overwriting. Major concerns include insufficient novelty (perceived as incremental over prior prototype works), lack of robustness analysis for the confusion matrix against pseudo-label noise, inadequate evaluation of computational overhead and hyperparameter sensitivity, and insufficient justification for design choices like the thresholding strategy. These raise concerns about the paper's technical soundness and contribution level.

**Reviewer Concerns:**

The rebuttal partially addresses concerns about hyperparameter sensitivity and computational overhead by providing additional ablation tables and timing comparisons. It also clarifies the source of ground truth in Figure 2 and the symmetry of the confusion matrix. However, core concerns remain largely outstanding. The rebuttal does not convincingly elevate the perceived incremental novelty or provide a rigorous, standalone analysis of the confusion matrix's robustness to error accumulation and CAM noise over many steps. The defense of the thresholding and weighting strategies, while providing ablations, is not fundamentally strengthened.

**Reviewer Scores:**

Based on discussion, it is unlikely the reviewers would have significantly raised their scores. Reviewers, who explicitly questioned the fundamental contribution and robustness, would probably maintain their negative stance. The rebuttal might have slightly alleviated some minor clarity issues, but without tackling the key concerns about the paper's core innovation and soundness, an overall score increase may be improbable.

---

### Decision · Program_Chairs · 2026-01-26

Reject